# KITS: INDUCTIVE SPATIO-TEMPORAL KRIGING WITH INCREMENT TRAINING STRATEGY

## ABSTRACT

Sensors are commonly deployed to perceive the environment. However, due to the high cost, sensors are usually sparsely deployed. Kriging is the tailored task to infer the unobserved nodes (without sensors) using the observed source nodes (with sensors). The essence of kriging task is transferability. Recently, several inductive spatio-temporal kriging methods have been proposed based on graph neural networks, being trained based on a graph built on top of observed nodes via pretext tasks such as masking nodes out and reconstructing them. However, the graph in training is inevitably much sparser than the graph in inference that includes all the observed and unobserved nodes. The learned pattern cannot be well generalized for inference, denoted as *graph gap*. To address this issue, we first present a novel *Increment* training strategy: instead of masking nodes (and reconstructing them), we add virtual nodes into the training graph so as to mitigate the graph gap issue naturally. Nevertheless, the empty-shell virtual nodes without labels could have bad-learned features and lack supervision signals. To solve these issues, we pair each virtual node with its most similar observed node and fuse their features together; to enhance the supervision signal, we construct reliable pseudo labels for virtual nodes. As a result, the learned pattern of virtual nodes could be safely transferred to real unobserved nodes for reliable kriging. We name our new Kriging model with Increment Training Strategy as KITS. Extensive experiments demonstrate that KITS consistently outperforms existing kriging methods by large margins, e.g., the improvement over MAE score could be as high as **18.33%**.

## 1 INTRODUCTION

Sensors play essential roles in various fields like traffic monitoring (Zhou et al., 2021), energy control (Liu et al., 2022), and air quality monitoring (Cini et al., 2021). For example, in smart transportation, loop detectors are installed on roads to perceive traffic dynamics, such as vehicle flows and speeds. Nevertheless, due to the high cost of devices and maintenance expenses (Liang et al., 2019), the actual sparsely deployed sensors are far from sufficient to support various services that require fine-grained data. To address this problem, **Inductive Spatio-Temporal Kriging** (Appleby et al., 2020; Wu et al., 2021a; Hu et al., 2021; Zhang et al., 2022; Wu et al., 2021b; Zheng et al., 2023) is proposed to estimate values of *unobserved nodes* (without sensors) by using values of *observed nodes* (with sensors) across time.

The common settings of inductive kriging are: (1) the training is only based on observed nodes; (2) when there are new unobserved nodes inserted during inference, the model can naturally transfer to them without re-training (Wu et al., 2021a), i.e., being inductive. To enable such transferability, existing inductive methods mainly adopt the following training strategy (illustrated in Figure 1(a)): it constructs a graph structure on top of observed nodes (e.g., based on the spatial proximity of the nodes (Barthélemy, 2011; Mao et al., 2022)), randomly masks some observed nodes' values, and then trains a model (which is mostly Graph Neural Network (GNN) based) to reconstruct each node's value. In this strategy, the graph structure is commonly used to capture node correlations and the inductive GNNs such as GraphSAGE (Hamilton et al., 2017) accommodate different graph structures during inference, as shown in Figure 1(b), where the values of new nodes 4-5 will be inferred by the model trained from nodes 1-3. We name this strategy as **Decrement training strategy** since it *decrements* the number of observed nodes during training (by masking their values) and use them to mimic the unobserved nodes to be encountered during the inference phase.

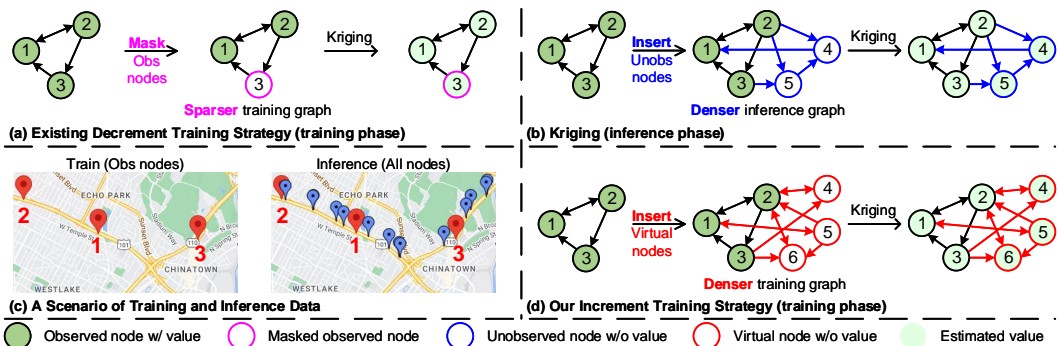

Figure 1: Decrement and Increment training strategies. (a) Decrement training strategy: observe nodes 1-3 during training, and **mask** node-3 out to reconstruct. (b) Kriging (inference phase): observe nodes 1-3, infer the values of new nodes 4-5. (c) A scenario of training and inference data. (d) Increment training strategy: observe nodes 1-3, **insert** virtual nodes 4-6 to mimic the target unobserved nodes in inference, and learn to directly estimate their values.

Unfortunately, the Decrement training strategy inevitably suffers from the *graph gap* issue, namely, the graph adopted for training is much sparser than that for inference, and it will aggravate with more nodes unobserved. Expressly, the *training graph* is based only on observed nodes (masked and unmasked), whereas the *inference graph* would be based on observed and unobserved nodes. There would be a clear gap between the two graphs: (1) the latter would have more nodes than the former; and (2) the two graph topologies would be different (e.g., the latter could be denser). This graph gap would pose a challenge to transfer from the training graph to the inference graph. For example, as a scenario shown in Figure 1(c), where red pins represent observed nodes, and blue pins represent unobserved nodes, the training graph (based on only red pins nodes) would significantly differ from the inference graph (based on red and blue pins). As shown in our empirical studies in Appx B.1, the average node degree of the training graphs from the decrement methods like Wu et al. (2021a;b) can be more than 70% lower than the real inference graph.

To mitigate this issue, we propose a new training strategy, shown in Figure 1(d): (1) it inserts some empty-shell *virtual nodes* and obtains the corresponding expanded graph, (2) it then trains a (kriging) model based on the graph with the observed nodes' values as labels in a semi-supervised manner. With this strategy, the gap between the training and inference graphs would be naturally reduced since the observed nodes in the two graphs are the same, and the virtual nodes in the former mimic the unobserved nodes in the latter. To further close the graph gap, it inserts virtual nodes in different ways and generates various training graphs (for covering different inference graphs). We name this strategy as **Increment training strategy** since it *increments* the number of nodes of the graph during training. Empirical study in Appx B.1 shows our method narrows the degree difference to only 15%.

However, due to abundant labels for observed nodes and the absence of labels on virtual nodes, the Increment training strategy faces the *fitting* issue (shown in Appx B.2): it could easily get overfitting and underfitting on observed and virtual nodes, leading to high- and low-quality features, respectively. We present two solutions: Firstly, we design **Reference-based Feature Fusion** module to align each virtual node with its most similar observed node and vice versa, then fuse their features. As a result, (1) virtual nodes could improve their low-quality features with high-quality features from similar observed nodes; (2) observed nodes, affected by low-quality features, are less likely to get overfitting. Secondly, we present a **Node-aware Cycle Regulation** to provide reliable pseudo labels (Cascante-Bonilla et al., 2021) for virtual nodes so that they would be well-regulated. Overall, our contributions are threefold as below:

- We identify the *graph gap* issue of Decrement training strategy that has been adopted by existing inductive kriging methods, and propose a novel Increment training strategy.

- We further develop the Reference-based Feature Fusion module and the Node-aware Cycle Regulation for handling the *fitting* issue of the proposed strategy.

- We conduct extensive experiments on eight datasets of three types, showing that our proposed model outperforms existing kriging methods consistently by large margins.

## 2 RELATED WORK

Kriging is a widely used technique in geostatistics for spatial interpolation, which involves predicting the value of a variable at an unsampled location based on the observed values of the variable at nearby locations. Spatio-temporal kriging extends this technique to include the temporal dimension, enabling the estimation of values of unobserved locations at different times.

Matrix factorization is one of the most representative methods for spatio-temporal kriging (Bahadori et al., 2014; Zhou et al., 2012; Takeuchi et al., 2017; Li et al., 2020a;b; Deng et al., 2021; Lei et al., 2022). For example, GLTL Bahadori et al. (2014) takes an input tensor $\mathcal{X}^{\text{location} \times \text{time} \times \text{variables}}$ with unobserved locations set to zero and then uses tensor completion to recover the values at the observed locations. GE-GAN (Xu et al., 2020) is another method, which builds a graph on top of both observed and unobserved nodes and utilizes node embeddings (Yan et al., 2006) to pair each unobserved node with the most relevant observed nodes for kriging. It then uses generative models (Goodfellow et al., 2020) to generate values for unobserved nodes. A more recent method GRIN (Cini et al., 2021) combines message passing mechanism (Gilmer et al., 2017) with GRU (Cho et al., 2014) to capture complex spatio-temporal patterns for kriging. These methods are limited by their "transductive" setting, i.e., they require the unobserved nodes to be known during the training phase: to handle new unobserved nodes that were not known before, they need model re-training.

More recently, quite a few methods, including KCN (Appleby et al., 2020), IGNNK (Wu et al., 2021a), LSJSTN(Hu et al., 2021), SpecKriging (Zhang et al., 2022), SATCN (Wu et al., 2021b), and INCREASE (Zheng et al., 2023) have been proposed to conduct spatio-temporal kriging in an "inductive setting" (which we call inductive spatio-temporal kriging). That is, during their training phase, the unobserved nodes are not known and they can handle new unobserved nodes without model re-training. These methods mainly adopt the Decrement training strategy: (1) it constructs a graph on top of the observed nodes, (2) it then randomly masks the values of some nodes of the constructed graph (which mimics the unobserved nodes), and (3) it then learns to recover the values of the unobserved nodes. However, as explained in Section 1, this Decrement training strategy would suffer from the *graph gap* issue, i.e., the training graph is based on all observed nodes, while the inference graph is based on both observed and unobserved nodes. In this paper, we propose a new *Increment training strategy*, which inserts virtual nodes to the training graph so as to mitigate the *graph gap* issue - with this strategy, the training graph is based on observed nodes and virtual nodes (which mimic unobserved nodes).

## 3 KITS: KRIGING WITH INCREMENT TRAINING STRATEGY

### 3.1 PROBLEM DEFINITION AND OVERVIEW OF KITS

**Problem Definition.** Let $\mathbf{X}^o_{T-t:T} \in \mathbb{R}^{N_o \times t}$ denote the values of $N_o$ observed nodes in $t$ time intervals. We follow existing studies (Wu et al., 2021a) and construct a graph structure on the observed nodes (e.g., creating an edge between two nodes if they are close enough). We denote the adjacency matrix of the graph structure by $\mathbf{A}^o \in [0,1]^{N_o \times N_o}$. The **inductive spatio-temporal kriging** problem is to estimate the values of $N_u$ unobserved nodes, which are not known until inference, based on $\mathbf{X}^o_{T-t:T}$ and the graph on top of the observed nodes and unobserved nodes.

**Overview of KITS.** To mitigate the graph issue suffered by existing methods, we first propose a new **Increment training strategy**, which inserts *virtual nodes* to the training graph and aims to estimate the values of all nodes with a kriging model in a semi-supervised manner (Section 3.2). We then design a kriging model with an encoder-decoder architecture, which involves two components, namely **Spatio-Temporal Graph Convolution** (STGC) and **Reference-based Feature Fusion** (RFF) module (Section 3.3). Finally, we incorporate a **Node-aware Cycle Regulation** (NCR) for regulating those virtual nodes since they lack of labels (Section 3.4). We name the whole framework **KITS** and provide its overview in Figure 2.

### 3.2 INCREMENT TRAINING STRATEGY FOR KRIGING

To mitigate the graph gap issue suffered by the existing Decrement training strategy, we propose a new **Increment training strategy**: It first inserts some empty-shell *virtual nodes* and obtains the

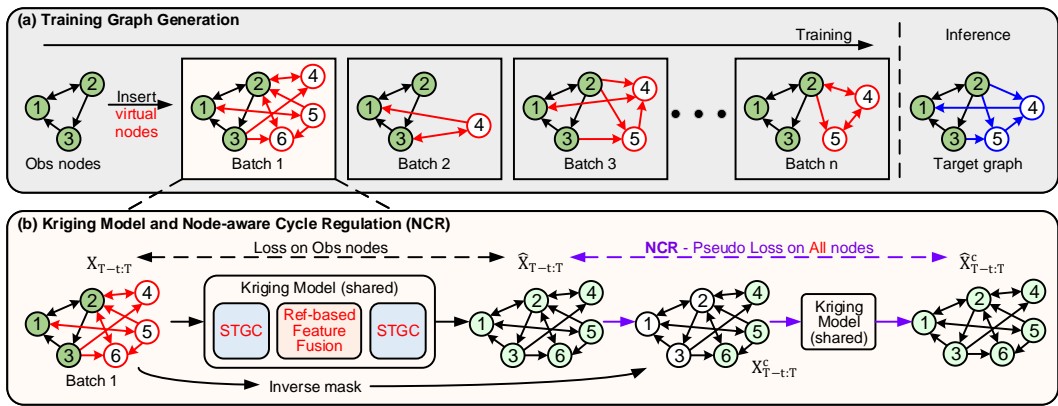

Figure 2: Overview of KITS. (a) Illustration of the procedure of generating multiple training graphs by inserting virtual nodes with randomness (so as to cover different possible inference graphs); (b) Illustration of the kriging model and the Node-aware Cycle Regulation (NCR) (based on Batch 1).

corresponding expanded graph, and then trains a (kriging) model based on the graph with the values of the observed nodes as labels in a semi-supervised manner (see Figure 1(d) for an illustration).

The core procedure of this training strategy is to insert some "virtual" nodes in the training graph to mimic the "unobserved" nodes in the inference graph, yet the unobserved nodes are not known during training. To implement this procedure, two questions need to be answered: (1) how many virtual nodes should be inserted; (2) how to create edges among observed nodes and virtual nodes.

Our solution about virtual nodes is as follows. First, we assume the availability of some rough estimate of the unobserved nodes to be encountered during inference by following existing studies (Wu et al., 2021a; Hu et al., 2021). That is, we assume the availability of a *missing ratio* $\alpha$, the ratio of unobserved nodes over all nodes in the inference graph. We denote $N_o$ as the number of observed nodes. Then, we insert $N_v$ virtual nodes, where $N_v = N - N_o = \frac{N_o}{1-\alpha} - N_o = \frac{\alpha \cdot N_o}{1-\alpha}$. Considering inference graphs with varying numbers of unobserved nodes, we add a random noise $\epsilon$ to $\alpha$. Besides, the values of the virtual nodes are initialized as 0, a common practice for nodes without readings.

To answer the second question about virtual edges, we adopt the following graph augmentation method. For each virtual node, we (1) pick an observed node randomly, (2) create an edge between the virtual node and the picked node; and (3) create an edge between the virtual node and each *neighbor node* of the picked node with a probability $p \sim Uniform[0, 1]$. The rationale is: the virtual node is created to mimic an unobserved node, which should have relations with a local neighborhood of observed nodes, e.g., a sensor has relations with others within a certain spatial proximity.

In addition, we generate multiple batches of training graphs and train batch by batch. Due to the randomness of the above procedure of inserting virtual nodes, we will generate various training graphs yet similar to the inference graph to some extent in different batches (see Figure 2(a) for illustration). This diversity will help to achieve better generality of the model to handle different inference graphs. A detailed description of the Increment training strategy is provided in the Appx C.

### 3.3 KRIGING MODEL

#### 3.3.1 SPATIO-TEMPORAL GRAPH CONVOLUTION (STGC)

STGC acts as the basic building block of the kriging model, and is responsible for aggregating spatio-temporal features from neighboring nodes (e.g., nearby nodes) to the current node with graph convolution (Cini et al., 2021). Specifically, we denote the input features of STGC as $\mathbf{Z}_i \in \mathbb{R}^{N \times D}$, where the subscript $i$ means the time interval is $T_i$, $N = N_o + N_v$ represents the total number of observed and virtual nodes, and $D$ is feature dimension. We have the following designs in STGC. First, to aggregate the features across different time intervals, for features $\mathbf{Z}_i$, we directly concatenate it with the features in the previous and following $m$ time intervals and denote the concatenated features as $\mathbf{Z}_{i-m:i+m} \in \mathbb{R}^{N \times (2m+1)D}$. Second, we aggregate the features across different nodes

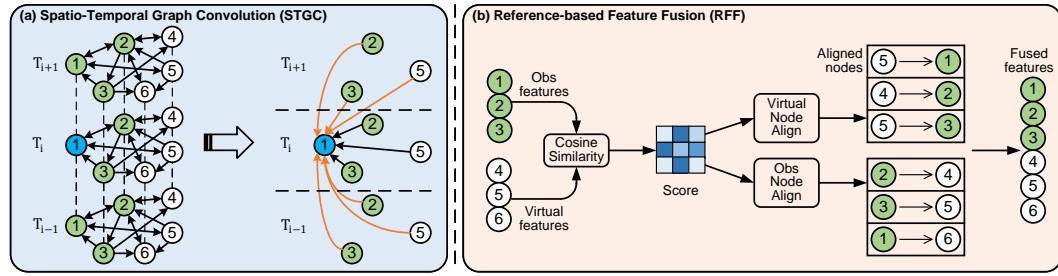

Figure 3: Details of (a) Spatio-Temporal Graph Convolution (STGC), take the data from three time intervals, and node-1 as an example, and (b) Reference-based Feature Fusion (RFF).

based on the training graph (indicated by the adjacency matrix $\mathbf{A} \in \mathbb{R}^{N \times N}$). Yet we prevent from aggregating features for a node from itself by masking the diagonal elements of adjacency matrix $\mathbf{A}$ (i.e., we remove the self-loops in the graph), denoted as $\mathbf{A}^-$. The rationale is that in spatio-temporal kriging, observed nodes have values in all time intervals, while virtual nodes have values missing in all time intervals. As a result, the observed/virtual nodes would have high/low-quality features all the time. In the case we allow each node to aggregate features from itself, the observed/virtual nodes would learn to improve their high/low-quality features with high/low-quality features, and thus the gap between their features quality would be widened, which would aggravate the overfitting/underfitting issues of observed/virtual nodes as mentioned in Section 1. An illustration of the STGC is shown in Figure 3(a). Formally, STGC can be written as:

$$\mathbf{Z}_i^{(l+1)} = FC(GC(\mathbf{Z}_{i-m:i+m}^{(l)}, \mathbf{A}^-)) \tag{1}$$

where $(l)$ and $(l+1)$ represent the layer indices, $FC(\cdot)$ is a fully-connected layer, and $GC(\cdot)$ is an inductive graph convolution layer (Cini et al., 2021).

### 3.3.2 REFERENCE-BASED FEATURE FUSION (RFF)

As mentioned earlier, there exists a quality gap between observed nodes' features and the virtual nodes' ones. To deal with the gap, we propose a **Reference-based Feature Fusion** (RFF) module, which pairs observed nodes and virtual nodes and then fuses their features. The rationale is that for a virtual node, its low-quality features would be improved with the high-quality features of its paired observed node - this would help to mitigate the underfitting issue of the virtual node; and for an observed node, its high-quality features would be affected by the low-quality features of its paired virtual node - this would help to mitigate the overfitting issue of the observed node.

The RFF module, shown in Figure 3(b), works as follows. First, we calculate a similarity matrix $\mathbf{M}_s \in [0,1]^{N_o \times N_v}$, where $\mathbf{M}_{s,[i,j]}$ is the re-scaled cosine similarity score between the $i^{th}$ observed node and the $j^{th}$ virtual node based on their features. Second, we apply the $\arg\max$ operation to each row/column of $\mathbf{M}_s$ to obtain an index vector $\mathbf{Ind}^*$ and an similarity vector $\mathbf{S}^*$, which indicate the most similar observed/virtual node of each virtual/observed node. Third, we pair each observed/virtual node to its most similar virtual/observed node based on the index vector $\mathbf{Ind}^*$. Fourth, we fuse the features of the nodes that are paired with each other with a shared FC layer and re-scale them based on the similarity vector $\mathbf{S}^*$. The procedure (for the $i^{th}$ virtual node) is given as:

$$\mathbf{Z}_i^v = FC(\mathbf{Z}_i^v || \mathbf{S}_i^* \odot Align(\mathcal{N}^o, \mathbf{Ind}_i^*)) \tag{2}$$

where $Align(\cdot)$ extracts the features of the most similar observed node according to its index $\mathbf{Ind}_i^*$, $\odot$ is the element-wise matrix multiplication, and $\mathcal{N}^o$ denotes the set of observed nodes. Some evidences, showing the effectiveness of RFF for the fitting issue, are provided in Appx B.2.

### 3.4 NODE-AWARE CYCLE REGULATION (NCR)

Recall that virtual nodes do not have supervision signals in the training process. Therefore, we propose to construct pseudo labels to better regulate the learning on virtual nodes. Specifically, we propose **Node-aware Cycle Regulation** (NCR) (as illustrated in Figure 2(b)), which works as follows. We first conduct the kriging process once (first stage) and obtain the estimated values of

all nodes. We then swap the roles of observed nodes and virtual nodes with an *inverse mask* and conduct the kriging process again (second stage) with the estimated values (outputted by the first stage kriging process) as *pseudo labels*. The key intuition is that during the second stage kriging process the virtual nodes would have supervision signals (i.e., pseudo labels) for regulation. We note that similar cycle regulation techniques have been used for traffic imputation tasks (Xu et al., 2022), and our NCR differs from existing techniques in that it uses an *inverse mask* but not a *random mask*, i.e., it is node aware and more suitable for kriging problem. Formally, NCR can be written as:

$$\hat{\mathbf{X}}_{T-t:T} = KM(X_{T-t:T}, \mathbf{A}^-) \tag{3}$$

$$\mathbf{X}^c_{T-t:T} = (\mathbf{1} - \mathbf{M}_{T-t:T}) \odot \hat{\mathbf{X}}_{T-t:T} \tag{4}$$

$$\hat{\mathbf{X}}^c_{T-t:T} = KM(\mathbf{X}^c_{T-t:T}, \mathbf{A}^-) \tag{5}$$

where $\mathbf{X}_{T-t:T}$ is the input data, $KM(\cdot)$ is the kriging model, $\hat{\mathbf{X}}_{T-t:T}$ is the output of the kriging model (first stage), $(\mathbf{1}-\mathbf{M}_{T-t:T})$ is the inverse mask, and $\hat{\mathbf{X}}^c_{T-t:T}$ is the output of the kriging model (second stage). Finally, the overall loss function could be written as:

$$\mathcal{L} = MAE(\hat{\mathbf{X}}_{T-t:T}, \mathbf{X}_{T-t:T}, \mathbf{I}_{obs}) + \lambda \cdot MAE(\hat{\mathbf{X}}^c_{T-t:T}, \hat{\mathbf{X}}_{T-t:T}, \mathbf{I}_{all}) \tag{6}$$

where $MAE(\cdot)$ is mean absolute error, $\mathbf{I}_{obs}$ and $\mathbf{I}_{all}$ mean calculating losses on observed and all nodes, respectively, and $\lambda$ is a hyperparameter controlling the importance of pseudo labels.

## 4 EXPERIMENTS

We firstly introduce the experimental settings in Section 4.1. Then, we make comparisons with the state-of-the-art kriging models in Section 4.2. Finally, we show the ablation studies in Section 4.3.

### 4.1 EXPERIMENTAL SETTINGS

**Datasets.** We use 4 datasets in traffic, 2 in air quality, and 2 in solar power (details in Appx D).

Table 1: Summary of 8 datasets.

| Datasets | Traffic Speed | | | Traffic Flow | Air Quality (PM2.5) | | Solar Power | |
|---|---|---|---|---|---|---|---|---|
| | METR-LA | PEMS-BAY | SEA-LOOP | PEMS07 | AQI-36 | AQI | NREL-AL | NREL-MD |
| Region | Los Angeles | San Francisco | Seattle | California | Beijing | 43 cities in China | Alabama | Maryland |
| Nodes | 207 | 325 | 323 | 883 | 36 | 437 | 137 | 80 |
| Timesteps | 34,272 | 52,128 | 8,064 | 28,224 | 8,759 | 8,760 | 105,120 | 105,120 |
| Granularity | 5min | 5min | 5min | 5min | 1hour | 1hour | 5min | 5min |
| Start time | 3/1/2012 | 1/1/2017 | 1/1/2015 | 5/1/2017 | 5/1/2014 | 5/1/2014 | 1/1/2016 | 1/1/2016 |
| Data partition | Train/Val/Test: 7/1/2 | | | | Same as GRIN | | Train/Val/Test: 7/1/2 | |

**Baselines.** (1) Inductive kriging: Mean imputation, OKriging (Cressie & Wikle, 2015), K-nearest neighbors (KNN), KCN (Appleby et al., 2020), IGNNK (Wu et al., 2021a), LSJSTN (Hu et al., 2021) and INCREASE (Zheng et al., 2023); (2) Transductive kriging: GLTL (Bahadori et al., 2014), MPGRU (Cini et al., 2021) and GRIN (Cini et al., 2021). More details are provided in Appx E.

**Evaluation metrics.** We mainly adopt Mean Absolute Error (MAE), Mean Absolute Percentage Error (MAPE) and Mean Relative Error (MRE) (Cini et al., 2021) as the evaluation metrics. We also include some results of Root Mean Square Error (RMSE) and R-Square (R2) in Appx F.

### 4.2 MAIN RESULTS

Apart from the inductive setting that we target in this paper, we consider the transductive setting to cover a broad range of settings of spatio-temporal kriging. The difference between these two settings is that in the former, the unobserved nodes are not available for training while in the latter, they are available. Our KITS can be applied in the transductive setting by replacing the virtual nodes with actual unobserved nodes. We set the missing ratios $\alpha = 50\%$ for all datasets and present the results in Table 2 for inductive setting, and in Table 3 for transductive setting. Besides, the error bars evaluation is provided in Appx H.2.

**Inductive** kriging comparisons. Table 2 shows that our KITS consistently achieves state-of-the-art performance, e.g., KITS outperforms the best baseline by **18.33%** on MAE (AQI dataset) and **30.54%** on MAPE (NREL-MD). We attribute it to the fact that the proposed KITS has taken afore-mentioned *graph gap* and *fitting* issues into consideration: (1) we apply the novel Increment training strategy for kriging to benefit from diverse and dense training graphs. (2) utilizes a well-designed STGC module and another RFF module to improve the features quality of virtual (unobserved) nodes; (3) further presents an NCR to create reliable pseudo labels for kriging.

Table 2: Comparisons with **inductive** kriging baselines. "-" means some methods require the input of GPS coordinates or full distance information, which is not available in SEA-LOOP and PEMS07 datasets. The best results are shown in **bold**, and the second best are underlined. "Improvements" show the improvement of our KITS over the best baseline.

| Method | Traffic Speed | | | | | | | | | Traffic Flow | | |
| --- | --- | --- | --- | --- | --- | --- | --- | --- | --- | --- | --- | --- |
| | METR-LA (207) | | | PEMS-BAY (325) | | | SEA-LOOP (323) | | | PEMS07 (883) | | |
| | MAE | MAPE | MRE | MAE | MAPE | MRE | MAE | MAPE | MRE | MAE | MAPE | MRE |
| Mean | 8.6687 | 0.2534 | 0.1515 | 4.7068 | 0.1135 | 0.0752 | 6.0264 | 0.1794 | 0.1045 | 103.5164 | 0.9802 | 0.3380 |
| OKriging | 8.1886 | 0.2391 | 0.1431 | 4.7006 | 0.1131 | 0.0751 | - | - | - | - | - | - |
| KNN | 8.3068 | 0.2274 | 0.1452 | 4.4898 | 0.1000 | 0.0718 | 6.5510 | 0.1842 | 0.1136 | 109.4717 | 0.9425 | 0.3575 |
| KCN | 7.5972 | 0.2341 | 0.1326 | 5.7177 | 0.1404 | 0.0914 | - | - | - | - | - | - |
| IGNKK | 6.8571 | 0.2050 | 0.1197 | 3.7919 | 0.0852 | 0.0606 | 5.1865 | 0.1370 | 0.0901 | 80.7719 | 0.9314 | 0.2635 |
| LSJSTN | 7.0666 | 0.2066 | 0.1234 | 3.8609 | 0.0836 | 0.0617 | - | - | - | 101.7706 | 0.8500 | 0.3325 |
| INCREASE | 6.9096 | 0.1941 | 0.1206 | 3.8870 | 0.0835 | 0.0621 | 4.8537 | 0.1267 | 0.0842 | 93.7737 | 1.1683 | 0.3062 |
| KITS (Ours) | **6.1276** | **0.1714** | **0.1071** | **3.5911** | **0.0819** | **0.0574** | **4.2313** | **0.1141** | **0.0734** | **75.1927** | **0.6847** | **0.2456** |
| Improvements | **10.64%** | **11.70%** | **10.53%** | **5.30%** | **1.92%** | **5.28%** | **12.82%** | **9.94%** | **12.83%** | **6.91%** | **19.45%** | **6.79%** |

| Method | Air Quality | | | | | | Solar Power | | | | | |
| --- | --- | --- | --- | --- | --- | --- | --- | --- | --- | --- | --- | --- |
| | AQI-36 (36) | | | AQI (437) | | | NREL-AL (137) | | | NREL-MD (80) | | |
| | MAE | MAPE | MRE | MAE | MAPE | MRE | MAE | MAPE | MRE | MAE | MAPE | MRE |
| Mean | 20.9346 | 0.6298 | 0.2905 | 39.0240 | 1.4258 | 0.5973 | 4.5494 | 1.6164 | 0.3664 | 12.2784 | 3.9319 | 0.6927 |
| KNN | 18.2021 | 0.5130 | 0.2526 | 23.1718 | 0.7376 | 0.3547 | 4.4118 | 1.2381 | 0.3554 | 12.5239 | 3.3277 | 0.7066 |
| KCN | 20.6381 | 0.6190 | 0.2896 | 21.9771 | 0.6207 | 0.3278 | 4.6349 | 2.0685 | 0.3733 | 11.5863 | 4.5994 | 0.6537 |
| IGNKK | 22.3862 | 0.7892 | 0.3141 | 22.3997 | 0.7200 | 0.3341 | 2.1939 | 1.7267 | 0.1767 | 4.3950 | 2.4813 | 0.2480 |
| LSJSTN | 22.7715 | 0.9022 | 0.3209 | 20.1396 | 0.5314 | 0.3003 | 2.0827 | 1.0960 | 0.1677 | 4.3206 | 1.9160 | 0.2436 |
| INCREASE | 22.9031 | 1.0682 | 0.3214 | 19.9140 | 0.6130 | 0.2970 | 2.0936 | 1.1342 | 0.1686 | 4.7302 | 1.8029 | 0.2669 |
| KITS (Ours) | **16.5892** | **0.3873** | **0.2302** | **16.2632** | **0.4187** | **0.2489** | **1.8315** | **0.7812** | **0.1475** | **4.1181** | **1.2523** | **0.2323** |
| Improvements | **8.86%** | **24.50%** | **8.87%** | **18.33%** | **21.21%** | **16.20%** | **12.06%** | **28.72%** | **12.05%** | **4.69%** | **30.54%** | **4.64%** |

**Transductive** kriging comparisons. As mentioned before, we also test our method in a transductive setting, where the target nodes and the full graph structure are **known** during the training phase, which is easier than the inductive setting that this paper focuses on. Both settings have their application scenarios in practice. Table 3 shows similar conclusions: KITS improves MAE by **20.61%** on PEMS07 dataset and MAPE by up to **45.50%** on NREL-MD dataset. Even with unobserved node topology revealed in training (no more graph gap), the transductive methods still cannot beat KITS, since STGC, RFF, NCR modules better fuse features and offer pseudo supervision.

Table 3: Comparisons with **transductive** kriging baselines.

| Method | Traffic Speed | | | | | | | | | Traffic Flow | | |
| --- | --- | --- | --- | --- | --- | --- | --- | --- | --- | --- | --- | --- |
| | METR-LA (207) | | | PEMS-BAY (325) | | | SEA-LOOP (323) | | | PEMS07 (883) | | |
| | MAE | MAPE | MRE | MAE | MAPE | MRE | MAE | MAPE | MRE | MAE | MAPE | MRE |
| GLTL | 8.6372 | 0.2627 | 0.1508 | 4.6986 | 0.1128 | 0.0741 | - | - | - | - | - | |
| MPGRU | 6.9793 | 0.2223 | 0.1220 | 3.8799 | 0.0951 | 0.0620 | 4.6203 | 0.1393 | 0.0802 | 97.2821 | 1.1475 | 0.3177 |
| GRIN | 6.6096 | 0.1959 | 0.1155 | 3.8322 | 0.0845 | 0.0613 | 4.2466 | 0.1262 | 0.0743 | 95.9157 | 0.6844 | 0.3132 |
| KITS (Ours) | **6.0604** | **0.1708** | **0.1059** | **3.5809** | **0.0788** | **0.0572** | **4.1773** | **0.1132** | **0.0725** | **76.1451** | **0.6673** | **0.2487** |
| Improvements | **8.31%** | **12.81%** | **8.31%** | **6.56%** | **6.75%** | **6.69%** | **1.63%** | **10.30%** | **2.42%** | **20.61%** | **2.50%** | **20.59%** |

| Method | Air Quality | | | | | | Solar Power | | | | | |
| --- | --- | --- | --- | --- | --- | --- | --- | --- | --- | --- | --- | --- |
| | AQI-36 (36) | | | AQI (437) | | | NREL-AL (137) | | | NREL-MD (80) | | |
| | MAE | MAPE | MRE | MAE | MAPE | MRE | MAE | MAPE | MRE | MAE | MAPE | MRE |
| GLTL | 21.8970 | 0.6643 | 0.3073 | 30.9248 | 1.0133 | 0.4612 | 4.8230 | 1.5635 | 0.3885 | 12.4229 | 3.6326 | 0.7009 |
| MPGRU | 22.5312 | 0.9439 | 0.3126 | 22.7233 | 0.8289 | 0.3478 | 2.4439 | 1.8900 | 0.1968 | 5.3504 | 3.7190 | 0.3018 |
| GRIN | 16.7497 | 0.5235 | 0.2324 | 17.4716 | 0.5615 | 0.2674 | 1.9623 | 1.2376 | 0.1581 | 5.0114 | 2.5790 | 0.2827 |
| KITS (Ours) | **16.3307** | **0.3559** | **0.2266** | **16.2431** | **0.4097** | **0.2486** | **1.8090** | **0.6661** | **0.1465** | **4.1088** | **1.4056** | **0.2318** |
| Improvements | **2.50%** | **32.02%** | **2.50%** | **7.03%** | **27.03%** | **7.03%** | **7.81%** | **46.18%** | **7.34%** | **18.01%** | **45.50%** | **18.00%** |

## 4.3 ABLATION STUDY

We conduct ablation studies to answer the following questions. **Q1:** Which module contributes the most to our method? **Q2:** How do Increment and Decrement training strategies behave on different

Table 4: Different ways of node insertion on METR-LA ($\alpha = 50\%$).

| Method | MAE | MAPE | MRE |
|--------|-----|------|-----|
| w/ all nodes | | | |
| Random | 6.4553 | 0.1886 | 0.1128 |
| w/ first-order neighbors | | | |
| $p=1$ | 6.4254 | 0.1817 | 0.1123 |
| $p=0.75$ | 6.4397 | 0.1863 | 0.1125 |
| $p=0.5$ | 6.4094 | 0.1854 | 0.1120 |
| $p=0.25$ | 6.4670 | 0.1860 | 0.1130 |
| $p=0$ | 6.8282 | 0.2089 | 0.1193 |
| $p=$random | **6.3754** | **0.1806** | **0.1114** |

Table 5: Component-wise ablation study on METR-LA ($\alpha = 50\%$). Column "INC" means whether Increment training strategy (✓) is used.

| Method | INC | NCR | STGC | RFF | MAE | MAPE | MRE |
|--------|-----|-----|------|-----|-----|------|-----|
| M-0 | | | | | 6.7597 | 0.1949 | 0.1181 |
| M-1 | ✓ | | | | 6.3754 | 0.1806 | 0.1114 |
| M-2 | ✓ | ✓ | | | 6.2607 | 0.1791 | 0.1094 |
| M-3 | ✓ | | ✓ | | 6.2107 | 0.1738 | 0.1117 |
| M-4 | ✓ | | | ✓ | 6.3269 | 0.1810 | 0.1106 |
| M-5 | ✓ | ✓ | ✓ | ✓ | **6.1276** | **0.1714** | **0.1071** |

missing ratios? **Q3:** Will the virtual nodes bring similar impacts as the real unobserved node? **Q4:** How different missing patterns affect the results?

Unless specified, we uniformly conduct ablation study on METR-LA dataset under the inductive setting, and with the missing ratio $\alpha = 50\%$. We include more experiments in Appx H.

**Different ways of node insertion (Q1(1)).** The first row (Random) of Table 4 means the newly-inserted virtual nodes are randomly connected to known nodes (observed nodes and inserted virtual nodes). In this case, the virtual node might connect to distant nodes. Several existing studies (Gilmer et al., 2017; Li et al., 2017; Ishiguro et al., 2019; Pham et al., 2017) have adopted this strategy for graph augmentation. Other rows randomly connect a virtual node to a known node, as well as a fraction $p$ of its first-order neighbors (this is the strategy adopted in this paper). According to the results, (1) our strategy of creating edges between a virtual nodes and a chosen node's neighbors works better than the commonly used strategy of creating edges based on all nodes and (2) it works better to use a random $p$ since it would generate more diverse training graphs.

**Ablation study on different modules (Q1(2)).** Table 5 validates the effectiveness of each proposed module. (1) We first compare between M-0 (Decrement training strategy) and M-1 (Increment training strategy), which share the same GCN model (Cini et al., 2021). The results show that Increment training strategy outperforms Decrement training strategy by a large margin (e.g., **5.69%** on MAE). (2) We then compare M-2, M-3, and M-4 with M-1. The results verify the benefit of each of the NCR, STGC and RFF modules. (3) The full model achieves the lowest MAE 6.1276, which is **9.37%** lower than that of M-0, which demonstrates the effectiveness of our proposed components.

**Baselines with different missing ratios (Q2(1)).** We compare KITS with other baselines with missing ratios $\alpha$ increasing from 20% to 80%, which means the kriging task becomes more challenging. Figure 4 shows that: (1) KITS consistently outperforms other baselines in all situations, which demonstrates the superiority and stability of our proposed modules; (2) The curves of baselines are steep, which means they are sensitive to missing ratios; and (3) When $\alpha$ is large, KITS benefits more from Increment training strategy and outperforms other inductive baselines with larger margins.

**Training strategies with different missing ratios (Q2(2) and Q3).** To further fairly compare the two different training strategies' behaviors under different missing ratios, we create a decrement version of our model: we change model M-5 in Table 5 with standard Decrement training strategy (Wu et al., 2021a) and the rest modules remain untouched. We use METR-LA and vary missing ratios $\alpha$ from 50% to 80%. (1) Red v.s. Green in Figure 5: With the increase of $\alpha$, increment one (red)'s advantage margin becomes larger over the decrement one (green). Since the *graph gap* issue becomes severe, the Decrement training strategy's performance deteriorates faster. **Virtual nodes have similar positive impacts as the real nodes (Q3)**: (2) Red v.s. Blue in Figure 5: We conduct a transductive version of KITS, which replaces the virtual nodes with real unobserved nodes (without values) for comparison. With all $\alpha$, the increment strategy could achieve similar results to the transductive setting, demonstrating that the created training graphs could achieve similar improvement as the real full graphs with unobserved nodes. Before $\alpha$ hits 76%, the difference of our virtual graph (red line, increment inductive) and the real graph (blue line, transductive) is highlighted with the light-blue region, which is only around 1.10% MAE; when $\alpha$ is larger than 76%, our virtual graph can offer even better performance than the real graph, highlighted in the light-red region.

**Different missing patterns (Q4).** So far, we focus on the *Random Missing* pattern (Figure 6(b)), where the observed nodes from the complete dataset (Figure 6(a)) are missed randomly. In this

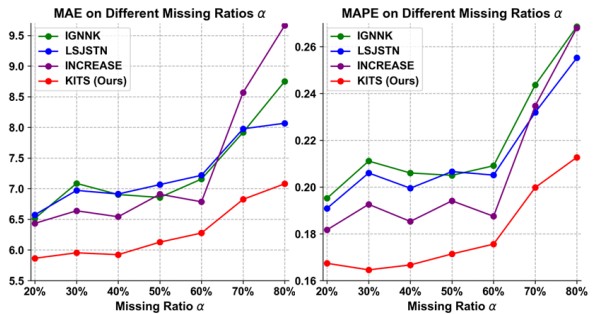

Figure 4: Comparisons among methods with different missing ratios $\alpha$.

Figure 5: Comparisons between Decrement and Increment training strategies.

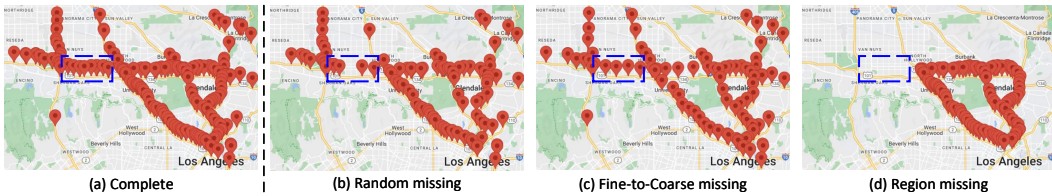

(a) Complete      (b) Random missing      (c) Fine-to-Coarse missing      (d) Region missing

Figure 6: Different missing patterns on METR-LA dataset. (a) Complete. (b) Random missing. (c) Fine-to-Coarse missing. (d) Region missing.

part, we consider two additional missing patterns, namely *Fine-to-Coarse Missing* (Figure 6(c)) and *Region Missing* (Figure 6(d)), which can be witnessed in practice as well. (1) *Fine-to-Coarse Missing* means that the nodes are missed in a regular manner, e.g., this happens when sensors are installed for every a certain amount of in-between distance. (2) *Region Missing* happens when sensors are installed on a small region as an early-phase test field and those in other regions are missing. The blue dotted rectangles could highlight the differences among the three missing patterns. Usually, *Region Missing* has the largest graph gap, and *Fine-to-Coarse Missing* has the least.

Table 6: Study on different missing patterns.

| Method | Random Missing | | | Fine-to-Coarse Missing | | | Region Missing | | |
|---|---|---|---|---|---|---|---|---|---|
| | MAE | MAPE | MRE | MAE | MAPE | MRE | MAE | MAPE | MRE |
| IGNKK | 6.8571 | 0.2050 | 0.1197 | 6.7903 | 0.2136 | 0.1180 | 8.6884 | 0.2761 | 0.1511 |
| LSJSTN | 7.0666 | 0.2066 | 0.1234 | 6.9758 | 0.2068 | 0.1213 | 8.6082 | 0.2659 | 0.1498 |
| INCREASE | 6.9096 | 0.1941 | 0.1206 | 6.7077 | 0.1853 | 0.1165 | 10.1537 | 0.2763 | 0.1766 |
| KITS (Ours) | **6.1276** | **0.1714** | **0.1071** | **5.6111** | **0.1568** | **0.0974** | **7.7676** | **0.2522** | **0.1350** |
| Improvements | **10.64%** | **11.70%** | **10.53%** | **16.35%** | **15.38%** | **16.39%** | **9.77%** | **5.15%** | **9.88%** |

According to Table 6, *Fine-to-Coarse Missing* is the easiest case (with the lowest error). *Region Missing* is the hardest case (with the highest error) because in *Region Missing*, most unobserved nodes do not have nearby observed nodes' values as references, which are essential for GNN baselines to learn high-quality features, thus causing the baselines' fitting problems and bad performances. KITS still displays its strong inductive ability due to RFF and NCR to tackle fitting issues.

## 5 CONCLUSION

In this paper, we study the inductive spatio-temporal kriging problem. We first show that existing methods mainly adopt the Decrement training strategy, which would cause a gap between training and inference graphs (called the *graph gap* issue). To mitigate the issue, we propose a new *Increment training strategy*, which inserts *virtual nodes* in the training graph to mimic the unobserved nodes in the inference graph, so that the gap between the two graphs would be reduced. We further design two modules, namely Reference-base Feature Fusion (RFF) and Node-aware Cycle Regulation (NCR), for addressing the fitting issues caused by the lack of labels for virtual nodes. We finally conduct extensive experiments on eight datasets, which consistently demonstrate the superiority of our proposed model.

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

## Appendix

## A    SOURCE CODE

The source code, as well as the adopted datasets, are provided in the attached .zip file.

## B    EVIDENCES FOR ISSUES

### B.1    GRAPH GAP ISSUE

Table 7: Empirical statistics for graph gap issue.

| Graph Source | Avg degree | Median degree | Min degree | Max degree |
|---|---|---|---|---|
| IGNNK's Training Graphs | 19.351 | 20 | 13 | 21 |
| KITS's Training Graphs | 28.627 | 28 | 21 | 43 |
| Inference Graph | 33 | 33 | 33 | 33 |

To show some direct evidence of the graph gap issue, we run our KITS (with increment training strategy) and IGNNK (with decrement training strategy) on METR-LA ($\alpha = 50\%$) dataset for 1,000 batches, and show the avg/median/min/max *largest degree* of 1,000 training graphs in Table 7. We could observe that (1) the largest node degree of IGNNK is never as large as that of the inference graph (21 v.s. 33); and (2) our KITS can mitigate this issue well, as the average largest degree of 1,000 training graphs is close to that of the inference graph (28.627 v.s. 33).

### B.2    FITTING ISSUE

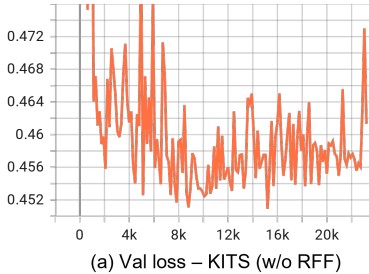
(a) Val loss – KITS (w/o RFF)

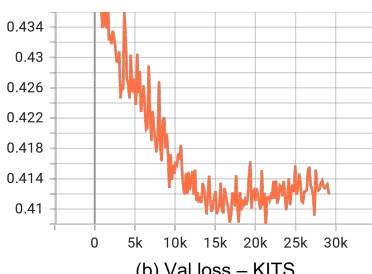
(b) Val loss – KITS

Figure 7: Validation losses of (a) KITS (w/o RFF) and (b) KITS. The loss is calculated based on the unobserved nodes after each training epoch (some iterations).

For the fitting issue, we show a direct evidence related to the Reference-based Feature Fusion (RFF) module, which is dedicated to dealing with gap of the high-quality observed nodes and low-quality virtual nodes. We plot the validation losses (calculated on target unobserved nodes after each training batch) of KITS and KITS (w/o RFF) in Figure 7, and could draw the following conclusions:

(1) Both (a) KITS (w/o RFF) and (b) KITS suffer from the fitting issue, e.g., the validation losses start increasing after some iterations;

(2) The proposed RFF module helps to mitigate the issue, because: the validation loss curve of KITS is much smoother than that of KITS (w/o RFF); KITS gets overfitting after 16k iterations, while KITS (w/o RFF) gets overfitting quickly after 8k iterations; and the validation loss of KITS (0.41) is much better than that of KITS (w/o RFF, 0.45), e.g., 8.89% better.

## C PSEUDO CODE FOR INCREMENT TRAINING STRATEGY FOR KRIGING

In Section 3.2, we describe the way of inserting virtual nodes. We further provide the pseudo code in Algorithm 1. Note that the insertion is batch-wise, and we only show the code to generate graphs, but omit other training and optimization details.

---

**Algorithm 1** Increment Training Strategy for Kriging

---

**Input:** adjacency matrix of observed nodes $A^o$, missing ratio $\alpha$, number of observed nodes $N_o$, observed nodes' first-order neighbors map $\mathcal{M}^o$

**Output:** adjacency matrix $A$ for observed and virtual nodes

1: **for** $epoch = 1, 2, \ldots$ **do**
2:     **for** $batch = 1, 2, \ldots$ **do**
3:         $N_v \leftarrow int(N_o/(1 - \alpha + \epsilon)) - N_o$     ▷ Estimate virtual node number, $\epsilon \in [0, 0.2]$
4:         $N \leftarrow N_o + N_v$     ▷ Total number of nodes
5:         $A \leftarrow [0, 1]^{N \times N}$     ▷ Initialize full adjacency matrix
6:         $A[: N_o, : N_o] \leftarrow A^o$
7:         $\mathbf{M} \leftarrow \{0\}^{N \times N}$     ▷ Indicator matrix for masking $A$
8:         $\mathbf{M}[: N_o, : N_o] = 1$
9:         **for** $e = N_o + 1, N_o + 2, \ldots, N_o + N_v$ **do**     ▷ Insert virtual nodes in sequence
10:             $(v, \mathcal{N}_v) \leftarrow \mathcal{M}^o$     ▷ Randomly select a node and its neighbors
11:             $\mathcal{C} \leftarrow \{v\}$     ▷ Initialize the connected candidates
12:             Generate a random probability $p \in [0, 1]$
13:             Add $p$ percent $\mathcal{N}_v$ to $\mathcal{C}$ to be candidates
14:             Add $(e, \mathcal{C})$ to $\mathcal{M}^o$     ▷ Add first-order neighbors of current virtual node
15:             **for** $c \in \mathcal{C}$ **do**
16:                 Add $e$ to $\mathcal{M}^o[c]$     ▷ Update first-order neighbors
17:                 Randomly select a status from $\{0, 1, 2\}$     ▷ 0 - forward; 1 - backward; 2 - both
18:                 **if** 0 or 2 **then**
19:                     $\mathbf{M}[c, e] = 1$
20:                 **end if**
21:                 **if** 1 or 2 **then**
22:                     $\mathbf{M}[e, c] = 1$
23:                 **end if**
24:             **end for**
25:         **end for**
26:         $A \leftarrow A \odot \mathbf{M}$     ▷ Adjacency matrix for observed and virtual nodes
27:     **end for**
28: **end for**

---

## D DATASETS

**Basic description.** We conduct experiments on the following 8 datasets: (1) *METR-LA* [1] is a traffic speed dataset that contains average vehicle speeds of 207 detectors in Los Angeles County Highway. The time period is from 3/1/2012 to 6/27/2012, and the sample rate is 5 minutes. (2) *PEMS-BAY* [1] is another popular traffic speed dataset, which contains 325 sensors installed in Bay Area, San Francisco. The sample rate is also 5 minutes, and the data is from 1/1/2017 to 6/30/2017. (3) *SEA-LOOP* [2] contains the speed readings of 2015 from 323 inductive loop detectors, which are deployed in the Greater Seattle Area, the default sample rate is 5 minutes. (4) *PEMS07* [3], also known as PEMSD7, is a traffic flow dataset with 883 sensors and the sample rate is 5 minutes, and the period is from 5/1/2017 to 8/31/2017. This dataset is constructed from Caltrans Performance Measurement System in California. (5) *AQI-36* [4] is a reduced version of AQI that is selected from the Urban Computing Project (Zheng et al., 2014). Existing kriging works only work on this reduced

---

[1] https://github.com/liyaguang/DCRNN
[2] https://github.com/zhiyongc/Seattle-Loop-Data
[3] https://github.com/Davidham3/STSGCN
[4] https://github.com/Graph-Machine-Learning-Group/grin

version that starts from 5/1/2014. (6) *AQI* [4] is short for Air Quality Index, and it contains the hourly measurements from 437 air quality monitoring stations, which are located at 43 cities in China. AQI contains the measurements of 6 pollutants, and following existing works like GRIN (Cini et al., 2021), we only study PM2.5. (7) *NREL-AL* [5], also known as Solar-Energy dataset, records the solar power outputted by 137 photovoltaic power plants in Alabama in 2016, and the sample rate is 5 minutes. (8) *NREL-MD* [5] is another solar energy dataset that has 80 photovoltaic power plants in Maryland.

**Data partition.** For all datasets except AQI-36 and AQI, we do not shuffle the data, and uniformly use 7/1/2 to split the complete datasets into train/validation/test sets. For AQI-36 and AQI, we follow GRIN (Cini et al., 2021) to take data from March, June, September and December to form the test set.

**Creating random missing.** For most of the conducted experiments, the missing ratio $\alpha$ is fixed as 50% for all datasets. We generate random masks to split all nodes into observed and unobserved nodes (which are used only for evaluation, and will not be used during training). Specifically, suppose there are $N$ nodes, and the missing ratio $\alpha = 50\%(0.5)$. Then, we randomly sample a vector from uniform distribution $U[0,1]^N$ to generate a random number for each node. Finally, we take the nodes whose value is less than $\alpha$ as unobserved nodes for inference. For reproducibility, we uniformly fix the random seed as 1 (of *numpy* and *random* libraries) for all the datasets by default.

**Data normalization.** Two common ways of data normalization are min-max normalization (MinMaxScaler of scikit-learn [6]), and zero-mean normalization (StandardScaler of scikit-learn).

Among all the datasets, PEMS07 is the only dataset where the data distributions of the test set is quite different from those of the training and validation set. Usually, zero-mean normalization will calculate the *mean* and *standard deviation (std)* values from the observed nodes in the training set, and use them to uniformly normalize all of the train, validation and test sets. However, due to the distributions differences, i.e., the *mean* and *std* values of the training set is quite different from those of the test set, it is not a good choice to use zero-mean normalization here. Fortunately, for flow data, the minimum (usually 0) and maximum flow values could be easily determined. Thus, we use min-max normalization for PEMS07. In addition, for NREL-AL and NREL-MD, the capacity (maximum value) of each node (plant) could be directly obtained. Therefore, we follow IGNNK (Wu et al., 2021a) and LSJSTN (Hu et al., 2021) to use min-max normalization for these two datasets (each node will be normalized with their own capacities, i.e., maximum values).

For all other datasets, zero-mean normalization is uniformly adopted.

**Constructing adjacency matrix.** One of the most common ways of constructing adjacency matrix $A$ among nodes is using a thresholded Gaussian kernel to connect a node to nearby nodes within a specific radius. The formula is provided as follows.

$$A_{i,j} = \begin{cases} exp(-\frac{dist(i,j)^2}{\gamma}), & dist(i,j) \leq \delta \\ 0, & \text{otherwise} \end{cases} \tag{7}$$

where $A$ is the adjacency matrix, $i$ and $j$ are the indices of two nodes, $dist$ measures the geographical distances between nodes, $\gamma$ represents kernel width, and $\delta$ is the threshold (radius).

For the datasets adopted for experiments, we categorize them into three groups:

(1) *METR-LA, PEMS-BAY, PEMS07*: For these datasets, the connectivity among nodes are provided in the form of (node 1, node 2, distance), which could directly replace the condition of Equation 7.

(2) *SEA-LOOP*: The adjacency matrix is directly provided, but there is no geographic coordinates information about nodes (which are essential to some baselines).

(3) *AQI-36, AQI, NREL-AL, NREL-MD*: In these datasets, the geographic coordinates of nodes are offered, and we directly follow Equation 7 to calculate the adjacency matrix. We follow GRIN (Cini et al., 2021) to set $\gamma$ as the standard deviation of distances among nodes.

---

[5] https://www.nrel.gov/grid/solar-power-data.html
[6] https://scikit-learn.org/

# E  BASELINES

The selected baselines are: (1) Inductive kriging: Mean imputation, OKriging [7] (Cressie & Wikle, 2015), K-nearest neighbors (KNN), KCN [8] (Appleby et al., 2020), IGNNK [9] (Wu et al., 2021a), LSJSTN [10] (Hu et al., 2021) and INCREASE [11] (Zheng et al., 2023); (2) Transductive kriging: GLTL [9] (Bahadori et al., 2014), MPGRU [12] (Cini et al., 2021) and GRIN [12] (Cini et al., 2021). We provide a bit more details of some baselines as follows.

To begin with, we explain the adaptions we made to IGNNK, LSJSTN and INCREASE. They originally do not perform standard validation (on valid set) after each training epoch, but directly validate on the test set (for inference), and save the model whose test score is the best. To make fair comparisons, we adapt them to uniformly perform standard validation (on valid set) after each training epoch, and save the model with the best validation score for testing (inference).

**GLTL (Bahadori et al., 2014).** It is short for Greedy Low-rank Tensor Learning, and is a transductive spatio-temporal kriging method. It takes an input tensor of size $location \times time \times variables$ with unobserved locations set to zero and then uses tensor completion to recover the values at the observed locations.

**MPGRU and GRIN (Cini et al., 2021).** GRIN originally targets the data imputation problem, and to the best to our knowledge, it is the only imputation baseline that covers the kriging task. MPGRU is a reduced version of GRIN, which combines message passing mechanism with Gated Recurrent Unit (GRU) to capture complex spatio-temporal patterns for kriging.

**Mean imputation.** In data imputation task, a mean value is calculated for each node, based on its own available data from all time intervals. However, in spatio-temporal kriging, unobserved nodes do not have values all the time, i.e., calculating mean values for each node does not work. Instead, we calculate a mean value based on the available data (of all nodes) in each time interval.

**OKriging.** Ordinary Kriging takes advantage of the geographical information of nodes, and a Gaussian process to perform spatial interpolation.

**KNN.** K-Nearest Neighbors tries to use the average value of K nearest nodes to interpolate readings of unobserved nodes. We set $K = 10$ in all datasets.

**KCN (Appleby et al., 2020).** For each unobserved node, Kriging Convolutional Network firstly finds its K-nearest neighbors, and then utilizes a graph neural network to interpolate this node by aggregating the information of its neighbors. KCN is an inductive spatial interpolation method, and performs kriging for each time step independently. There are three variants of KCN: (1) KCN with graph convolutional networks; (2) KCN with graph attention networks; and (3) KCN with GraphSAGE. We select the last option in our experiments.

**IGNNK (Wu et al., 2021a).** It makes two modifications over KCN: (1) It considers temporal information; (2) The neighbors of a node are not restricted to observed nodes only. Then, IGNNK takes the spatio-temporal information of nearby observed and unobserved nodes to interpolate the value of current node.

**LSJSTN (Hu et al., 2021).** LSJSTN is similar to IGNNK, and it further improves the aggregation of temporal information by decoupling the model to learn short-term and long-term temporal patterns.

**INCREASE (Zheng et al., 2023).** INCREASE is a latest inductive spatio-temporal kriging method, and it is similar to KCN that aligns each node with K-nearest observed nodes. Besides, it tries to capture diverse spatial correlations among nodes, e.g., functionality correlation (calculated based on Point-Of-Interest (POI) data). Nevertheless, such POI information is usually unavailable, e.g., in our selected datasets, only one has such information. Therefore, we select the reduced version of INCREASE (w/o POI information) for comparisons.

---

[7] https://github.com/GeoStat-Framework/PyKrige
[8] https://github.com/tufts-ml/KCN
[9] https://github.com/Kaimaoge/IGNNK
[10] https://github.com/hjf1997/DualSTN
[11] https://github.com/zhengchuanpan/INCREASE
[12] https://github.com/Graph-Machine-Learning-Group/grin

## F   EVALUATION METRICS

We mainly adopt Mean Absolute Error (MAE), Mean Absolute Percentage Error (MAPE) and Mean Relative Error (MRE) (Cao et al., 2018; Cini et al., 2021) to evaluate the performance of all methods, and use Root Mean Square Error (RMSE) and R-Square (R2) in some experiments. Their formulas are written as follows.

$$MAE = \frac{1}{|\Omega|} \sum_{i \in \Omega} |\mathbf{Y}_i - \hat{\mathbf{Y}}_i| \tag{8}$$

$$MAPE = \frac{1}{|\Omega|} \sum_{i \in \Omega} \frac{|\mathbf{Y}_i - \hat{\mathbf{Y}}_i|}{|\mathbf{Y}_i|} \tag{9}$$

$$MRE = \frac{\sum_{i \in \Omega} |\mathbf{Y}_i - \hat{\mathbf{Y}}_i|}{\sum_{i \in \Omega} |\mathbf{Y}_i|} \tag{10}$$

$$RMSE = \sqrt{\frac{1}{|\Omega|} \sum_{i \in \Omega} (\mathbf{Y}_i - \hat{\mathbf{Y}}_i)^2} \tag{11}$$

$$R2 = 1 - \frac{\sum_{i \in \Omega} (\mathbf{Y}_i - \hat{\mathbf{Y}}_i)^2}{\sum_{i \in \Omega} (\bar{\mathbf{Y}} - \mathbf{Y}_i)} \tag{12}$$

where $\Omega$ is the index set of unobserved nodes for evaluation, $\mathbf{Y}$ is the ground truth data, $\hat{\mathbf{Y}}$ is the estimation generated by kriging models, $\bar{\mathbf{Y}}$ is the average value of labels.

## G   IMPLEMENTATION DETAILS FOR REPRODUCIBILITY

Our code is implemented with Python 3.8, PyTorch 1.8.1, PyTorch Lightning 1.4.0, and CUDA 11.1. All experiments are conducted on an NVIDIA Telsa V100, with 32 GB onboard GPU memory.

If not specified explicitly, we fix the *random seed* (of *numpy*, *random*, *PyTorch*, and *PyTorch Lightning* libraries) as 1, and the *missing ratio* $\alpha = 50\%$. For all datasets, we set the *window size* in the temporal dimension as 24, e.g., in Section 3.1, the $t$ in $\mathbf{X}_{T-t:T}^{obs}$ is 24. The *feature dimension* is set as 64. The *batch size* is set as 32. For the Incremental training strategy (Section 3.1), the *noise* $\epsilon$ is randomly chosen from $[0, 0.2]$ in each training batch, to make the number of virtual nodes vary. In STGC module (Section 3.2.1), $m$ is set as 1, i.e., we use the data from 1 historical, 1 current and 1 future time intervals to perform spatio-temporal features aggregation. In RFF module (Section 3.2.2), we align and fuse each observed/virtual node with *1 most similar virtual/observed node*. As for NCR (Section 3.3), in Equation 6, the parameter $\lambda$ is set to 1.

We utilize *Adam optimizer* for optimization, the *learning rate* is fixed as 0.0002, and use "CosineAnnealingLR" as the *learning rate scheduler*. In addition, we adopt *gradient normalization* (value=1.0) to make the training stable. The maximum *epoch* number is 300, and we utilize the *early stop mechanism*, say, we perform validation after each training epoch. If the validation performance has not been improved for 50 epochs, then we stop the training process. The model with the best validation performance is saved and used for testing (inference).

## H   ADDITIONAL EXPERIMENTAL RESULTS

### H.1   ADDITIONAL EVALUATION METRICS OF MAIN RESULTS

As existing inductive spatio-temporal kriging methods like IGNNK, LSJSTN and INCREASE adopt other metrics, including Root Mean Square Error (RMSE) and R-Square (R2), for inference (evaluation), we further provide their main results of 8 datasets in Table 8. We could find that the results of RMSE and R2 show similar clues as those of MAE, MAPE and MRE (as provided in the paper) do, e.g., our KITS could achieve 11.63% performance gain of RMSE in the AQI-36 dataset, and the improvement of R2 metric could reach 18.97% on AQI dataset, which could demonstrate the superiority of our proposed modules.

Table 8: Comparisons with **inductive** kriging baselines based on evaluation metrics of RMSE and R2. "-" means some methods require the input of GPS coordinates or full distance information, which is not available in SEA-LOOP dataset. The best results are shown in **bold**, and the second best are underlined. "Improvements" show the improvement of our KITS over the best baseline.

| Method | Traffic Speed | | | | | | Traffic Flow | |
|---|---|---|---|---|---|---|---|---|
| | METR-LA (207) | | PEMS-BAY (325) | | SEA-LOOP (323) | | PEMS07 (883) | |
| | RMSE | R2 | RMSE | R2 | RMSE | R2 | RMSE | R2 |
| IGNNK | 10.0701 | 0.2094 | **6.2607** | **0.5116** | 7.5498 | 0.5426 | 114.4463 | 0.3837 |
| LSJSTN | 10.4867 | 0.1456 | 6.5393 | 0.4637 | - | - | 142.9067 | 0.0394 |
| INCREASE | 10.6721 | 0.1116 | 6.7186 | 0.4374 | 7.5520 | 0.5387 | 130.8876 | 0.1928 |
| KITS (Ours) | **9.8636** | **0.2465** | 6.3438 | 0.4984 | **7.0652** | **0.5963** | **110.8695** | **0.4209** |
| Improvements | 2.05% | 17.72% | -1.33% | -2.58% | 6.42% | 9.90% | 3.13% | 9.70% |

| Method | Air Quality | | | | Solar Power | | | |
|---|---|---|---|---|---|---|---|---|
| | AQI-36 (36) | | AQI (437) | | NREL-AL (137) | | NREL-MD (80) | |
| | RMSE | R2 | RMSE | R2 | RMSE | R2 | RMSE | R2 |
| IGNNK | 35.9153 | 0.7075 | 33.4604 | 0.6001 | 3.3968 | 0.8621 | 7.2386 | 0.7774 |
| LSJSTN | 35.5725 | 0.7123 | 34.9258 | 0.5612 | 3.4176 | 0.8606 | 7.2541 | 0.7767 |
| INCREASE | 40.5021 | 0.6217 | 32.5495 | 0.6189 | 3.3258 | 0.8681 | 8.0543 | 0.7251 |
| KITS (Ours) | **31.4345** | **0.7882** | **29.4482** | **0.7363** | **3.0634** | **0.8881** | **7.0489** | **0.7895** |
| Improvements | 11.63% | 10.66% | 9.53% | 18.97% | 7.89% | 2.30% | 2.62% | 1.56% |

Figure 8: METR-LA datasets generated by different random seeds, with the missing ratio $\alpha = 50\%$. We regard random missing as a combination of fine-to-coarse missing and region missing, and the blue rectangles are an example showing that the patterns created by different random seeds are different. (a) Complete data. (b) Seed 0. (c) Seed 1. (d) Seed 2. (e) Seed 3. (f) Seed 4.

## H.2 ERROR BARS EVALUATION

Standard error bars evaluation means running each model for multiple times on the *same* dataset, and report the average value and standard deviation of each metric, which could show the learning stability of each model given randomness (e.g., model parameter initialization, etc.). Furthermore, recall that each adopted dataset has a missing ratio $\alpha$, which *randomly divides* the nodes (of each dataset) into observed nodes (for training), and unobserved nodes (for inference). In this section, we not only examine the learning ability (as standard error bars evaluation), but also check the stability of models given *different nodes divisions*, i.e., in each running, the observed and unobserved nodes would change. We run each method 5 times, and use random seeds 0-4 for this purpose. The created datasets are shown in Figure 8, and we could observe that the missing situations are quite different on different random seeds (e.g., the blue rectangles in the figure). We run each inductive kriging methods for each seed, and report their results in Table 9.

Table 9: Error bars of inductive methods with 5 different random seeds on METR-LA dataset. The best results are shown in **bold**, and the second best are underlined. "Improvements" show the improvement of our KITS over the best baseline.

| Method | Metric | Seed 0 | Seed 1 | Seed 2 | Seed 3 | Seed 4 | Mean ± Std |
|---|---|---|---|---|---|---|---|
| Mean | | 8.7155 | 8.6687 | 8.4157 | 9.0051 | 8.7997 | 8.7209±0.2138 |
| OKriging | | 8.3091 | 8.1886 | 8.0316 | 8.6314 | 8.2473 | 8.2816±0.2210 |
| KNN | | 8.7080 | 8.3068 | 8.5784 | 8.8114 | 8.5157 | 8.5841±0.1928 |
| KCN | | 7.9734 | 7.5972 | 8.1840 | 8.1049 | 7.9975 | 7.9714±0.2257 |
| IGNKK | MAE | 7.3065 | 6.8571 | 7.1895 | 7.3759 | 7.2739 | 7.2006±0.2034 |
| LSJSTN | | 7.4577 | 7.0666 | 7.3971 | 7.3200 | 7.5242 | 7.3531±0.1770 |
| INCREASE | | 7.1644 | 6.9096 | 7.5274 | 7.1350 | 7.4584 | 7.2390±0.2531 |
| KITS (Ours) | | **6.3230** | **6.1276** | **6.4373** | **6.1523** | **6.4415** | **6.2963±0.1507** |
| Improvements | | **11.74%** | **10.64%** | **10.46%** | **13.77%** | **11.44%** | **12.56%** |
| Mean | | 0.2510 | 0.2534 | 0.2593 | 0.2647 | 0.2703 | 0.2597±0.0080 |
| OKriging | | 0.2360 | 0.2391 | 0.2409 | 0.2537 | 0.2520 | 0.2443±0.0080 |
| KNN | | 0.2279 | 0.2274 | 0.2316 | 0.2367 | 0.2381 | 0.2323±0.0049 |
| KCN | | 0.2392 | 0.2341 | 0.2393 | 0.2540 | 0.2498 | 0.2433±0.0083 |
| IGNKK | MAPE | 0.2143 | 0.2050 | 0.2121 | 0.2251 | 0.2212 | 0.2155±0.0079 |
| LSJSTN | | 0.2097 | 0.2066 | 0.2133 | 0.2278 | 0.2206 | 0.2156±0.0086 |
| INCREASE | | 0.2013 | 0.1941 | 0.2027 | 0.2091 | 0.2069 | 0.2028±0.0058 |
| KITS (Ours) | | **0.1786** | **0.1714** | **0.1915** | **0.1791** | **0.1899** | **0.1821±0.0084** |
| Improvements | | **11.28%** | **11.70%** | **5.53%** | **14.35%** | **8.22%** | **10.21%** |
| Mean | | 0.1517 | 0.1515 | 0.1469 | 0.1562 | 0.1538 | 0.1520±0.0034 |
| OKriging | | 0.1446 | 0.1431 | 0.1402 | 0.1497 | 0.1442 | 0.1444±0.0034 |
| KNN | | 0.1516 | 0.1452 | 0.1497 | 0.1529 | 0.1489 | 0.1497±0.0029 |
| KCN | | 0.1389 | 0.1326 | 0.1425 | 0.1405 | 0.1395 | 0.1388±0.0037 |
| IGNKK | MRE | 0.1273 | 0.1197 | 0.1252 | 0.1279 | 0.1269 | 0.1254±0.0033 |
| LSJSTN | | 0.1300 | 0.1234 | 0.1289 | 0.1270 | 0.1313 | 0.1281±0.0031 |
| INCREASE | | 0.1248 | 0.1206 | 0.1311 | 0.1237 | 0.1301 | 0.1261±0.0044 |
| KITS (Ours) | | **0.1101** | **0.1071** | **0.1124** | **0.1067** | **0.1126** | **0.1098±0.0028** |
| Improvements | | **11.78%** | **10.53%** | **10.22%** | **13.74%** | **11.27%** | **12.44%** |

Table 10: More simplified error bars evaluation of selected inductive methods with 5 different random seeds on PEMS07, AQI-36 and NREL-AL datasets. The evaluation metric is MAE.

| Method | Dataset | Seed 0 | Seed 1 | Seed 2 | Seed 3 | Seed 4 | Mean ± Std |
|---|---|---|---|---|---|---|---|
| IGNKK | | 86.3293 | 80.7719 | 84.1235 | 83.6449 | 83.5836 | 83.6906±1.9801 |
| LSJSTN | | 105.6552 | 101.7706 | 107.6139 | 114.9543 | 115.1753 | 109.0339±5.8940 |
| INCREASE | PEMS07 | 97.8983 | 93.7737 | 101.4767 | 97.8384 | 97.6087 | 97.7192±2.7269 |
| KITS (Ours) | | **77.5165** | **75.1927** | **79.2799** | **77.1688** | **74.6181** | **76.7552±1.8797** |
| Improvements | | **10.21%** | **6.91%** | **5.76%** | **7.74%** | **10.73%** | **8.29%** |
| IGNKK | | 20.9982 | 22.3862 | 28.2143 | 22.3619 | 21.7623 | 23.1446±2.8900 |
| LSJSTN | | 18.7659 | 22.7715 | 26.7743 | 21.2729 | 19.5970 | 21.8363±3.1630 |
| INCREASE | AQI-36 | 18.1515 | 22.9031 | 35.7717 | 17.7529 | 18.0956 | 22.5350±7.6996 |
| KITS (Ours) | | **16.8937** | **16.5892** | **19.6877** | **16.9184** | **16.6375** | **17.3453±1.3177** |
| Improvements | | **6.93%** | **25.90%** | **26.47%** | **4.70%** | **8.06%** | **20.57%** |
| IGNKK | | 2.9220 | 2.1939 | 3.2298 | 2.8030 | 2.2087 | 2.6715±0.4566 |
| LSJSTN | | 2.1050 | 2.0827 | 2.5529 | 2.2206 | 2.1103 | 2.2143±0.1967 |
| INCREASE | NREL-AL | 2.7912 | 2.0936 | 3.1575 | 2.7772 | 2.2694 | 2.6178±0.4310 |
| KITS (Ours) | | **2.0920** | **1.8315** | **2.2962** | **1.9533** | **1.8843** | **2.0115±0.1867** |
| Improvements | | **0.62%** | **12.06%** | **10.06%** | **12.04%** | **10.71%** | **9.16%** |

From Table 9, we could observe that: (1) Our KITS consistently outperforms all other baselines by large margins on three metrics, namely, MAE, MAPE and MRE. For example, the average MAE score of KITS could surpass IGNNK by 12.56%, and our average MAPE score is 10.21% better than that of INCREASE. (2) KITS appears to be stable given different random missing situations, e.g., its standard deviation of MAE scores is 0.1507, while the second best is 0.1770. (3) The randomness raised by different parameter initialization seems to have minor impacts on the training of KITS.

More error bars evaluation on PEMS07, AQI-36 and NREL-AL datasets are provided in Table 10, where we only compare MAE scores of KITS with those of IGNNK, LSJSTN and INCREASE, and similar conclusions can be drawn.

## H.3    ABLATION STUDY ON SPATIO-TEMPORAL GRAPH CONVOLUTION (STGC)

STGC is introduced in Section 3.2.1, and in this section, we further conduct two groups of experiments on STGC (M-3 of Table 5 in the paper) as follows: (1) We claim that self-loop and temporal information from a node itself would aggravate the *fitting* issue (i.e., do harm to kriging) and therefore, we drop them. To support this claim, we conduct experiments to add them back and compare their results with the version without them; (2) There is a hyperparameter $m$, showing the amount of historical and future data, from which the STGC would aggregate spatio-temporal information. We also vary the number of $m$ to study its effects.

It could be observed from Table 11 that: (1) Aggregating temporal information from a node itself (e.g., integrate $T_{i-1}$'s features to $T_i$'s features for node-1) would cause a huge performance drop (the MAE score is dropped from 6.2107 to 6.7104, which is 8.05% worse), and adding self-loop makes it even worse, resulting in the MAE score dropping from 6.2107 to 6.9537. During training, observed nodes consistently have accurate labels, while virtual nodes do not, and this would cause to learn high- and low-quality features on observed nodes and virtual nodes, respectively. And the experimental results support our claim that self-loop and temporal information (from a node itself) would make the situation even worse. (2) In terms of the value of hyperparameter $m$, the best results are achieved when $m = 2$. Nevertheless, a larger $m$ means more GPU memories are required. After making trade-offs between GPU memory and scores, we set $m = 1$ in our experiments, whose results differ little from those of $m = 2$, but could save more memory cost.

Table 11: Ablation study on STGC module.

| Method | MAE | MAPE | MRE |
|---|---|---|---|
| STGC | **6.2107** | **0.1738** | **0.1117** |
| STGC + temporal | 6.7104 | 0.1993 | 0.1173 |
| STGC + self-loop | 6.9537 | 0.2176 | 0.1215 |
| $m = 0$ | 6.3754 | 0.1806 | 0.1114 |
| $m = 1$ | 6.2107 | 0.1738 | 0.1117 |
| $m = 2$ | **6.1905** | **0.1727** | **0.1082** |
| $m = 3$ | 6.2056 | 0.1768 | 0.1085 |
| $m = 4$ | 6.2101 | 0.1734 | 0.1085 |

Table 12: Ablation study on NCR. $\lambda$ is the hyperparameter that controls the importance of pseudo labels.

| Method | MAE | MAPE | MRE |
|---|---|---|---|
| $\lambda = 0.0$ | 6.3754 | 0.1806 | 0.1114 |
| $\lambda = 0.5$ | 6.2845 | 0.1795 | 0.1098 |
| $\lambda = 1.0$ | **6.2607** | **0.1791** | **0.1094** |
| $\lambda = 1.5$ | 6.4832 | 0.1921 | 0.1133 |
| $\lambda = 2.0$ | 6.5412 | 0.1958 | 0.1143 |

## H.4    ABLATION STUDY ON NODE-AWARE CYCLE REGULATION (NCR)

NCR is introduced in Section 3.4, which introduces a hyperparameter $\lambda$ in the loss function (Eq 6 in the paper) to control the significance of pseudo labels. We start with model M-2 (of Table 5 in the paper), and change the hyperparameter $\lambda$ to see their effects.

From Table 12, we could find that when $\lambda = 1$, the best results could be achieved. Performance drop could be noticed when $\lambda < 1.0$ or $\lambda > 1.0$, which means when $\lambda = 1$, the effects of observed nodes' real labels (first-stage learning) and NCR's pseudo labels (second-stage learning) reach a balance. In spite of the importance of introducing pseudo labels to regulate the learning process on virtual nodes, the pseudo labels created by NCR would never be as accurate as the real ones, hence, $\lambda$ should not be too large.

## H.5    ABLATION STUDY ON REFERENCE-BASED FEATURE FUSION (RFF)

Recall that the Increment training strategy suffers from the *fitting* issue that would generate high-quality and low-quality features for observed nodes and virtual nodes, respectively. By intuition, the most straightforward idea is to low down the features quality of observed nodes or improve the features quality of virtual node. Therefore, we further conduct experiments to study: (1) what information should be aligned for each node to adjust the quality of features; and (2) how much extra information should be used.

Table 13: Ablation study of different alignment ways on RFF module. "Obs" represents observed node, and "Vir" represents virtual node.

| Method | MAE | MAPE | MRE |
|---|---|---|---|
| Obs ← most similar Vir
Vir ← most similar Obs | **6.3269** | **0.1810** | **0.1106** |
| Obs ← most similar Obs
Vir ← most similar Obs | 6.4148 | 0.1849 | 0.1121 |
| Obs ← most dissimilar Obs
Vir ← most similar Obs | 6.3727 | 0.1784 | 0.1114 |

Table 14: Ablation study of the alignment number on RFF module. E.g., "Num Aligned=2" means aligning each observed/virtual node with 2 most similar virtual/observed nodes.

| Method | MAE | MAPE | MRE |
|---|---|---|---|
| Num Aligned=1 | **6.3269** | **0.1810** | **0.1106** |
| Num Aligned=2 | 6.3685 | 0.1845 | 0.1113 |
| Num Aligned=3 | 6.4253 | 0.1884 | 0.1123 |
| Num Aligned=all | 6.4310 | 0.1839 | 0.1124 |

For the first problem, as shown in Table 13, we consider three situations: (1) Align each observed/virtual node with its most similar virtual/observed node. In this case, the low-quality features of virtual node could benefit from high-quality features to improve its quality, while observed nodes, affected by virtual nodes, would be less likely to get overfitting; (2) Align each observed/virtual node with its most similar observed node. We aim to uniformly integrate current high-quality and low-quality features with low-quality features, so that their quality could both be improved; (3) Align each observed/virtual node with its most dissimilar/similar node. Different from (2), observed nodes could benefit from high-quality features, but the aligned high-features are not the best candidate. We aim to let observed node learn "slower" in this case.

In all situations, we uniformly provide a most similar observed node for each virtual node, and their differences lie in the information for observed nodes. From the table, we could know that using virtual nodes to affect observed nodes appear to be the best way to mitigate the *fitting* issue.

For the second problem, we propose to align each observed/virtual node with $\{1, 2, 3, all\}$ virtual/observed nodes and show the results in Table 14. We could draw the following conclusion: when the number is 1, best results could be obtained. Take (1) "Num Aligned=1" and (2) "Num Aligned=2" as an example: (1) All nodes uniformly learn a pattern of fusing "1 high- + 1 low-quality" features; (2) Observed nodes learn a pattern of fusing "1 high + 2 low", while virtual nodes learns another pattern of fusing "1 low + 2 high", which are different and have some conflicts. Because the model parameters are shared by all nodes, learning a uniform pattern could get better results than using multiple patterns. Therefore, we set the number as 1 in all our experiments.

## H.6 COMPARISONS OF INDUCTIVE ABILITIES

Table 15: Comparisons of inductive abilities among different kriging methods. Take "METR-LA → PEMS-BAY" as an example, the "origin" version of each method means training and inference on the same PEMS-BAY dataset, while "transfer" version means training on METR-LA dataset, and inference on PEMS-BAY dataset. "Decay Rate" shows the deterioration from "origin" to "transfer", which shows the inductive ability of each method. The best results are shown in **bold**, and the second best are underlined.

| Method | METR-LA → PEMS-BAY | | | NREL-MD → NREL-AL | | |
|---|---|---|---|---|---|---|
| | MAE | MAPE | MRE | MAE | MAPE | MRE |
| IGNNK (origin) | 3.7919 | 0.0852 | 0.0606 | 2.1939 | 1.7267 | 0.1767 |
| IGNNK (transfer) | 5.1586 | 0.1106 | 0.0825 | 2.2701 | 1.7095 | 0.1829 |
| Decay Rate | -36.04% | -29.81% | -36.14% | -3.47% | 1.00% | -3.51% |
| LSJSTN (origin) | 3.8609 | 0.0836 | 0.0617 | 2.0827 | 1.0960 | 0.1677 |
| LSJSTN (transfer) | 4.4635 | 0.0931 | 0.0713 | 2.3182 | 1.0863 | 0.1866 |
| Decay Rate | -15.61% | -11.36% | -15.56% | -11.31% | 0.89% | -11.27% |
| INCREASE (origin) | 3.8870 | 0.0835 | 0.0621 | 2.0936 | 1.1342 | 0.1686 |
| INCREASE (transfer) | 4.5482 | 0.0931 | 0.0727 | 2.4323 | 1.1867 | 0.1959 |
| Decay Rate | -17.01% | -11.50% | -17.07% | -16.18% | -4.63% | -16.19% |
| Ours (origin) | 3.5911 | 0.0819 | 0.0574 | 1.8315 | 0.7812 | 0.1475 |
| Ours (transfer) | 4.0153 | 0.0852 | 0.0642 | 1.8642 | 0.5894 | 0.1502 |
| Decay Rate | **-11.81%** | **-4.03%** | **-11.85%** | **-1.79%** | 24.55% | **-1.83%** |

Following IGNNK, we also conduct experiments to explore the inductive ability (i.e., transferability) of different kriging models. We select IGNNK, LSJSTN and INCREASE for comparison in this experiment. For each model, two versions are made: (1) *origin*: training and inference on the same target dataset; and (2) *transfer*: training on another dataset but inference on the target dataset. We select two target datasets, namely, PEMS-BAY and NREL-AL. For PEMS-BAY, the *transfer* version models are trained on METR-LA, while for NREL-AL, models are trained on NREL-MD. This setting is very practical, because usually we could only train a deep learning model on a small area, but need to apply them to other areas directly.

From Figure 15, we could know that: (1) All methods inevitably witness a performance drop when comparing *transfer* version to *origin* version; (2) Our KITS has the lowest decay rates on two datasets, and in particular, the MAPE score of *transfer* version KITS is 24.55% better than the *origin* version KITS, which strongly prove that KITS has excellent inductive ability.

## H.7 GENERALIZABILITY OF INCREMENT TRAINING STRATEGY

Table 16: Generalizability of Increment training strategy. "IGNNK" is the original version using Decrement training strategy, while "IGNNK + Increment" means employing our proposed Increment training strategy to IGNNK.

| Method | METR-LA (207) | | | AQI (437) | | |
|---|---|---|---|---|---|---|
| | MAE | MAPE | MRE | MAE | MAPE | MRE |
| IGNNK | 6.8571 | 0.2050 | 0.1197 | 22.3997 | 0.7200 | 0.3341 |
| IGNNK + Increment | 6.6077 | 0.1908 | 0.1154 | 20.5226 | 0.5984 | 0.3061 |
| Improvement | **3.64%** | **6.93%** | **3.59%** | **8.38%** | **16.89%** | **8.38%** |

Finally, we explore the generalizability of our proposed Increment training strategy. Specifically, we check if it could be directly applied to existing methods that use Decrement training strategy, and make some positive impacts.

We apply our strategy to IGNNK, and show the results in Table 16. On both METR-LA and AQI datasets, by changing from Decrement training strategy to Increment training strategy, i.e., we insert some virtual nodes (as well as the expanded graph structure) during training, the scores could be well improved, e.g., the MAE and MAPE score on AQI dataset is boosted by 8.38% and 16.89%, respectively. This could demonstrate the generalizability of our strategy.

## H.8 FURTHER DISCUSSION ON TRAINING STRATEGIES

We discuss an alternative training strategy for avoiding the graph gap issue, which we call the *Single training strategy* as follows. Suppose we have 100 observed nodes, and need to perform kriging for 100 unobserved nodes. For each training iteration, we would randomly mask 1 observed node, and use the remaining 99 observed nodes to reconstruct its value. During inference, the 100 unobserved nodes would not be inserted at a time, instead, they would be separately connected to 100 observed nodes and generate 100 inference graphs (i.e., we need to perform kriging 100 times). With each inference graph, we would use 100 observed nodes to estimate a value for the inserted unobserved node. With this training strategy, each training graph involves 99 observed nodes and 1 masked node and each inference graph involves 100 observed nodes and 1 unobserved node, and thus the gap between the training graphs and inference graphs is negligible.

Table 17: Kriging situations and MAE results of *Increment*, *Decrement* and *Single* training strategies. Suppose there are $x$ observed nodes and $x$ unobserved nodes in each dataset. "Iterations" represents the times of kriging (i.e., using kriging model) during inference.

| Strategy | #Training nodes | #Inference nodes | #Iterations | METR-LA (207) | SEA-LOOP (323) | PEMS07 (883) | AQI (437) |
|---|---|---|---|---|---|---|---|
| Single | $x$ | $x+1$ | $x$ | 7.3560 | 4.9766 | 146.2889 | 28.2472 |
| Decrement | $x$ | $2x$ | 1 | 6.7597 | 4.5732 | 86.7923 | 17.5867 |
| Increment | $2x$ | $2x$ | 1 | **6.3754** | **4.3688** | **82.9259** | **16.5797** |

We summarize the graph situations and kriging results of three training strategies, including Single, Decrement (existing) and Increment (ours), in Table 17. We could draw the following conclusions:

(1) In terms of the number of nodes in training and inference graphs, Single and our Increment training strategies could guarantee that they could be similar, i.e., the *graph gap* issue is addressed. However, there exists a clear gap for the Decrement training strategy; (2) Both Decrement and our Increment training strategies can krige all unobserved by running the trained kriging model once. Nevertheless, Single training strategy need to run $x$ times, which is inefficient; (3) Although the Single training strategy can avoid the graph gap, the relationships among unobserved nodes can never be used, which explains why its MAE scores are the worst; (4) Our proposed Increment training strategy performs the best in all aspects, including graph gap, efficiency, graph sparseness and MAE scores.

## I  LIMITATION AND FUTURE DIRECTION

We indicate the limitation of KITS, and provide an important future direction for kriging: (1) *Limitation.* By inserting virtual nodes during training, the training costs would inevitably be increased, e.g., when the missing ratio $\alpha = 50\%$, we would roughly double the number of (observed) nodes, and thus we double the training cost. (2) *Future direction.* In Table 6 (of the paper), KITS could still outperform other methods on the practical *Region Missing* case, but the scores are not good enough. This is because region missing is an extremely hard case in kriging, say, most of the unobserved nodes cannot have *nearby observed nodes* to help kriging. None of existing methods (including our KITS) has provided an effective solution for this case, and tackling this issue is an interesting research direction.

