# OpenReview forum: "KITS: Inductive Spatio-Temporal Kriging with Increment Training Strategy"
_ICLR.cc/2024/Conference — Submitted to ICLR 2024_

### Official Review · Reviewer_FRYk · 2023-10-27

**Soundness:** 1 poor
**Presentation:** 3 good
**Contribution:** 3 good
**Rating:** 5
**Confidence:** 4

**Summary:**

I've reviewed this paper for a previous conference and have discussed at length its merits and negative aspects with a long back and forth discussion with the authors. Unfortunately, none of my comments (and that of the other reviewers) were apparently taken into account as this submission is basically unchanged w.r.t. the previous iteration. Furthermore, several of the additional results presented during the previous discussion haven't been included in this version.

The paper introduces KITS, a novel approach for kriging based on graph neural networks. The main contribution of the paper is the introduction of an augmentation strategy based on the idea of adding virtual nodes to the input graph at training time. The augmentation strategy is then paired with a self-supervised training strategy yielding good results on several benchmark datasets. The method is presented as a solution to what the paper defines as the "graph gap", i.e., a mismatch between the graph at training and test time. Overall, the paper paper has merits, however, I do have some serious concerns that prevent me from recommending acceptance.

**Strengths:**

* The introduced data augmentation strategy paired with the self-supervised training routine is novel and appealing.
* Good empirical performance.
* Very good presentation.

**Weaknesses:**

* There is a conceptual flaw in the main motivation behind the introduced methodology. While it is straightforward to see why using a drastically different graph at training and inference time is a problem ("graph gap" in the paper), I do not understand why every target node should be reconstructed in a single forward pass.
* Reconstructing a single node at a time would remove the "graph gap", this would be the proper way of carrying out the evaluation.
* After removing nodes for training, graphs can become sparse. However, this issue is only caused by the removal of nodes for evaluation, i.e., it is an issue of the training/evaluation procedure and not an inherent issue of kriging methods. This would not be a problem in any real-world application as you would eventually train the model on the full graph.
* The considered datasets are quite small, removing many nodes at random might result in disconnected graphs that would explain the poor performance for some of the baselines.

I believe that the paper has good methodological novelty, although is not framed in the proper way as there is no inherent "graph gap" issues in spatio-temporal kriging. The paper should either be rewritten targeting a specific operational setting (heavily scaling back the current claims and significance of the work) or presenting the method as a data augmentation strategy on graphs that are not made artificially sparse.

**Questions:**

--

---

> ### Author Response · Authors · 2023-11-15
> **Response [1/3]**
>
> > **Summary**: Unfortunately, none of my comments (and that of the other reviewers) were apparently taken into account as this submission is basically unchanged w.r.t. the previous iteration.
>
> We would like to emphasize that **we have addressed all comments from the previous conference**, which we explain as follows.
> - **Reviewer 1 for the previous submission (weak accept - score 6)**:
>     - *W1*: This comment is that the reviewer did not observe significant improvement of our KITS over an existing method INCREASE due to some misinterpretation. We provided explanations on this in the rebuttal, with which the reviewer is satisfied. **No changes are necessary for this comment.**
>     - *W2*: This comment is on the robustness aspect of the paper. We provided explanations on this in the rebuttal, with which the reviewer is satisfied. **No changes are necessary for this comment.**
>     - *Q1, Q2*: The comments are some questions about the details of virtual node generation (Section 3.2). **We have re-written this part to make it clearer.**
>     - *Comment on our rebuttal*: **"I would like to thank the authors for their detailed and clear rebuttal. I am convinced with the provided answers. I keep my rating intact."** This comment shows that the reviewer is convinced with our responses to the comments in the rebuttal.
> - **Reviewer 2 for the previous submission (weak accept - score 6)**:
>     - *W1*: This comment suggests to include some empirical evidence about "graph gap issue" and "fitting" issue. **We have mentioned them in introduction and included the details in Appendix B.**
>     - *W2*: This comment suggests to include more error bars evaluation (like Table 9 in Appendix H.2). **We have followed this suggestion and further included Table 10 in Appendix H.2.**
>     - *W3, W4 (minor)*: These comments are about two typos in Equation 1 and 2. **We have fixed the typos.**
>     - *Comment on our rebuttal*: **"Thank you for your rebuttal! I updated my score as my concern raised in the review was resolved. I would recommend the discussion in the author's rebuttal be involved appropriately in the revised manuscript."** This comment shows that the reviewer is satisfied with our rebuttal overall. We have updated the draft as suggested by this reviewer.
> - **Reviewer 3 for the previous submission (you, borderline reject - score 4)**:
>     - *W1, W2, Q1*: Kriging single node at a time. **We have included and compared your suggested idea in Appendix H.8, which shows that the suggested idea does not work as well as our method for the problem settting we target in this paper.**
>     - *W3.1*: Isolated nodes introduced after dropping nodes. We replied to you with some statistics to answer this question during the rebuttal of the previous conference. **We believe this issue has been settled in our rebuttal, and thus there is no change for this comment.**
>     - *W3.2*: More discussions about region missing. **We have enriched the corresponding part, which is near Table 6.**
>     - *W5 (minor)*: More baselines such as SPIN by Marisca et al. NeurIPS 2022. We appreciated your comment, and replied to you with some experimental results about this baseline, but we finally decided not to include it for the following reasons. (1) This method targets transductive setting, which is not the main target setting of our paper; (2) It does not work as well as GRIN; (3) This method costed much in terms of GPU onboard memory, and we met out-of-memory (OOM) problem in PEMS07 dataset (with 883 nodes). In case that you still think we should include these results, we shall include them in a new version of our draft.
>     - *Q2*: Average degree of training and inference graphs. **We have included some statistics in Appendix B.1.**
>     - *Q3 (minor)*: Why removing self-loops in Equation 1. We have replied to you in the previous rebuttal.
>     - *Q4 (minor)*: Initialization of virtual nodes' representations. **We have re-written Section 3.2 to make it more clear.**
>     - *L1*: The possible negative effects of adding noise to training graphs. We previously replied to you with statistics in Figure 5.

---

> > ### Author Response · Authors · 2023-11-15
> > **Response [2/3]**
> >
> > - **Reviewer 4 for the previous submission (weak accept - score 6)**:
> >     - *Q1*: This comment is asking for clarifications on the input features of virtual nodes. **We have re-written Section 3.2 to make it clear.**
> >     - *Q2*: This comment suggests to consider *extra* information when creating graph edges, instead of using pure distance information among nodes. We agreed that such a way would possibly further improve the performace, however, by taking factors like fairness (with baselines) and the availability of extra information (e.g., POI information is not always available in our datasets) into consideration, **we decided to not introduce such extra information in our current work**.
> >     - *Q3*: This comment is about the lack of explanations about transductive setting (near Table 3). **We have included the differences between transductive and inductive settings in Section 4.2.**
> >     - *Q4*: This comment is about the impact of distribution shift in node-aware cycle regulation (Section 3.4). **We have convinced the reviewer our node-aware cycle regulation module would bring more benefits than negative impacts.**
> >     - *Q5*: This comment is asking for clarification on the parameter $p$ in Table 4. **We have re-written Section 3.2, and also established a connection between methodology and experiment part to make it clear.**
> >     - *Q6*: This comment suggests to visualize virtual nodes for better understanding. **We have convinced the reviewer that it's infeasible to do so with the current methodology (virtual nodes do not have GPS coordinates).**
> >     - *Comment on our rebuttal*: **"Thanks for putting together a nice rebuttal. Most of my questions have been answered. Some other comments: I know most of the GNN work assumes homophily and many real-world scenarios actually are, but some interesting observation is that when there is an accident, only a small area is (not even all neighbors) congested while if such value is missing, the method would tend to fail (I assume) as it relies too much on similarity and virtual nodes. This is not really a question but would be interesting to investigate in future work. I will raise my score to 6 accordingly."** This comment shows that the reviewer is satisfied with our rebuttal overall. We shall take the reviewer's additional suggestion for future work.
> >
> > > **W1**: Kriging single node at a time.
> >
> > We have included this suggested idea of training in Appendix H.8, which shows that the suggested idea does not work as well as our method for the problem settting we target in this paper (i.e., kriging on sparse graphs). We understand your point that when the training graph is dense enough, constructing single node at a time would have no graph gap issue. However, as we have replied to you previously, this setting is out of the scope of our paper, and we shall explore it as a future research direction.
> >
> > > **W1, W2, W3**: The sparsity of training graph.
> >
> > As mentioned in the first paragraph of introduction, the sensors are often **sparsely** deployed in real applications for saving cost. We refer to some datasets used in our experiments, which are both from real-world applications.
> > - PEMS07 comes from the Caltrans Performance Measurement System (PeMS) deployed in California (in use). It covers a large discrict, but the average node degree is 1.79 for the full graph (883 nodes), which means the graph is indeed locally sparse.
> > - Similarily, AQI-36 dataset is collected from an air quality monitoring application, where only 36 monitoring stations are installed for the whole Beijing City.
> >
> > In addition, many existing kriging studies assume sparsely distributed sensors, including IGNNK (Wu et al., 2021a), LSJSTN (Hu et al., 2021) and INCREASE (Zheng et al., 2023). We follow this line of research and target similar problem settings in our paper.

---

> > > ### Author Response · Authors · 2023-11-15
> > > **Response [3/3]**
> > >
> > > > **W4**: The considered datasets are quite small, removing many nodes at random might result in disconnected graphs that would explain the poor performance for some of the baselines.
> > >
> > > We collect some statistics of 4 datasets, including (1) the total number of nodes, (2) the number of isolated nodes (in the inference graph) and (3) the number of isolated nodes (in the training graph after dropping 50% nodes), as shown in the table below.
> > >
> > > |Dataset|#Node|#Isolated (inference)|#Isolated (train)|
> > > |-|-|-|-|
> > > |METR-LA|207|1|4|
> > > |PEMS-BAY|325|6|4|
> > > |SEA-LOOP|323|0|38|
> > > |PEMS07|883|27|135|
> > >
> > > Based on the statistics, we find that while dropping nodes for training, as adopted by baselines, might introduce some isolated nodes (more or less depending on the datasets), but the inferior performance of baselines cannot be majorly attributed by this phenomenon. This is because even in cases few or no isolated nodes are introduced (e.g., for the META-LA and PEMS-BAY datasets), their performance is inferior to ours by a clear margin (please refer to Table 2). We believe it is the graph gap issue that majorly affects the performance, which motivates us to propose our new increment training strategy to insert virtual nodes instead. In addition, we note that these datasets have been widely used in existing studies of spatio-temporal kriging.

---

> > > > ### Comment · Reviewer_FRYk · 2023-11-20
> > > >
> > > > As for our previous interactions, thank you for your comprehensive feedback.
> > > >
> > > > While I see your points and still find the paper methodologically significant, I keep my position regarding the "graph gap" issue. I do not think the mismatch between the graph used for training and evaluation is the relevant issue here as it can easily be avoided as previously discussed. The increment training strategy effectively acts as a data augmentation and graph rewiring routine, which can explain the improved performance in an inductive setting. This point is unchanged w.r.t. the previous version of the paper and, clearly, it seems that we cannot come to an agreement on this aspect.
> > > >
> > > > My position on the paper is unchanged, I believe that the introduced method is significant (the combination of the increment strategy with NCR is very appealing); however, I do not agree with your analysis of the above-mentioned aspects and think that the paper (and its impact on the research community) would benefit from a different framing. I'll keep my current score.

---

> > > > > ### Author Response · Authors · 2023-11-24
> > > > > **Thank you for your follow-up comments.**
> > > > >
> > > > > Thanks for recognizing the postive aspects of our work (i.e., "methodologically significant", "the introduced method is significant (the combination of the increment strategy with NCR is very appealing)", etc.). We would like to respond to your follow-up comments as follows.
> > > > >
> > > > > First, we would make our claims very concrete for ease of our discussions. We believe **we agree on the following claims**, but in case you would disagree with any of them, please let us know, and we shall discuss further.
> > > > >
> > > > > **Claim 1**: Existing studies (and their training strategies), including IGNNK (Wu et al., 2021a), LSJSTN (Hu et al., 2021) and INCREASE (Zheng et al., 2023), all suffer from the "graph gap" issue.
> > > > >
> > > > > **Claim 2**: Our KITS method proposed in this paper largely mitigates the "graph gap" issue.
> > > > >
> > > > > **Claim 3**: An alternative training method (which you suggested in the reivew of our previous submission) does not suffer from the "graph gap" issue. But as we have explained and also verified via experiments (Appendix H.8), it does not work as good as our KITS method for the setting targetted in this paper (i.e., when we have sparsely distributed sensors). It might work for some other settings (e.g., when we have densely distributed sensors), but they are out of the scope of our paper.
> > > > >
> > > > > In addition, we emphasize that we **did NOT claim** that the "graph gap" issue is one of the kriging task, as you have commented - *"I do not think the mismatch between the graph used for training and evaluation is the relevant issue here as it can easily be avoided as previously discussed."* We agree that it is not an inherent issue of the kriging task. We only claim that it is **an issue of existing inductive kriging methods (and their training strategies)** (i.e., the above Claim 1).
> > > > >
> > > > > Further, we agree that some graph augmentation elements are involved in our methodology (e.g., the way of inserting virtual nodes and constructing the edges is essentially a graph augmentation procedure). However, this graph augmentation procedure is only one of a few components of our methodology and there are alternative ways for the graph augmentation procedure. Therefore, we tend not to over emphasize the graph augmentation aspect of our methodology and frame it as a graph augmentation technique as you have suggested.
> > > > >
> > > > > Once again, we appreciate the comments you have made on our paper. We hope that you would give our response some further thoughts. In case you have further questions, please let us know and we will be more than happy to explain further. For your information, the author response period for our submissions has been extended until the end of December 1st. Thank you very much for your time and efforts on reviewing our paper and discussing with us!

---

> > > > > > ### Comment · Reviewer_FRYk · 2023-11-27
> > > > > >
> > > > > > As discussed several times, I believe the so-called "graph gap" issue is not a thing. Removing outgoing edges from the virtual nodes would effectively remove any mismatch between the training and target graph at inference time.
> > > > > >
> > > > > > As such, I believe that presenting the contributions of the paper as a mechanism to address such an issue is misleading and potentially harmful to future research on the topic. Conversely, I do believe that the proposed training (augmentation) strategy aligns well with the reconstruction (kriging) task, which can arguably explain the improvement in performance.
> > > > > >
> > > > > > The above justifies my current score.

---

### Official Review · Reviewer_env1 · 2023-11-03

**Soundness:** 3 good
**Presentation:** 2 fair
**Contribution:** 3 good
**Rating:** 6
**Confidence:** 3

**Summary:**

This paper addresses a spatio-temporal krigging problem. This is modeled as a node prediction task using graph convolution neural network (GCN) whose convolution operator is learned to compute the feature embeddings for each node, taking into account its topological connection to others (i.e., a training graph).

The learned weights of the convolution operator can be applied to any topological graphs that contain the training graph, thus enabling inductive inference on new node: a forward pass of the (learned) convolution over an expanded graph that includes incorporate new connections to unobserved nodes will predict the features for those nodes, which can be subsequently input to a feed-forward net for node prediction.

Previous approaches addressing this problem often trains the GCN on the observed (training) part of the entire graph, which is later used to perform inductive inference on the unobserved (test) part of the graph. Their performance might therefore suffered from the disparity or gap between the train and test graphs. This is what this paper aims to address.

This is based on three main ideas:

Graph Augmentation (Increment Strategy): Observed nodes are sampled. For each observed node, a virtual node and a connection to the observed node is created. The virtual node's connection to the neighborhood of the observed node is also randomly generated. Multiple such augmented graphs are generated to train the GCN so its learned convolution is robust to potential variation in the structure of the (unseen) test graph. Original features for the virtual nodes are set to be zero. Also, features of the same node reported or computed within a window of +/- m steps are also concatenated before passing through the GCN

Feature Fusion between Virtual & Observed Nodes: Generated features of virtual nodes and observed nodes from the above steps are paired based on a similarity notion. Paired features are concatenated and passed through a (learnable) neural net, which returns the fused features for the virtual node.

Pseudo-Label Generation: Pseudo-labels for virtual nodes are first generated using the learned GCN-based node prediction model. The model is then re-trained using only the pseudo-labels to predict labels of the observed nodes. The losses in two training phases can be combined to be optimized together.

**Strengths:**

The presented ideas are refreshingly interesting & novel to me.

The problem being addressed is also practically significant. The gap between the train and test graphs in the inductive setting of node prediction is always a fundamental issue, which needs an in-depth treatment.

The empirical studies are sufficiently extensive, showing good results across a variety of datasets.

Comprehensive ablation studies showing the effectiveness of each idea is also included.

**Weaknesses:**

Overall, I like this paper. It motivates well a fundamental issue of inductive inference with GCN.

But I do have a few concerns or questions regarding both the position, presentation and empirical evaluation of this paper that I want to discuss with the authors (mostly out of curiosity & for constructive feedback)

1. The main position of this paper is grounded in the spatio-temporal setting but the proposed treatment does not seem to have anything specific to the temporal aspect. The aggregation of temporal feature within +/- m steps is kind of a random treatment to me. I do not see a very particular reasoning why doing so makes sense. What if such aggregation erases important small-scale variation patterns in the time-series data? Furthermore, m is a value dependent on the nature of the data so even though the ablation studies conclude that m = 2 gives best result, it is still specific to the few set of experimental data and that should not be a generic guideline to choose m.

2. Although comparison with recent pure graph-based approaches has been well presented, I am still curious to know how well the proposed approach improves over a hybrid approach that integrates GCN with traditional krigging techniques. For example, the series of work on graph convolutional Gaussian processes. There is quite a substantial volume on that & I think the authors have not discussed that in the literature review. Such techniques can be used to address this krigging problem. After all, if I remember correctly, Gaussian processes were referred to as krigging in the past (in geo-statistics).

3. Last, in terms of the presentation, I believe it would be better if the authors spend some space on summarizing GCN which will highlight better the essence of transferability in the inductive setting of krigging. Otherwise, while the current presentation is fine for people who are familiar with graph neural network, it will still leave a lot of gap for generic readers.

**Questions:**

Based on the above, I have the following questions:

1. Could the authors position this work with the other literature on graph convolutional GPs, which is an integrative approach combining both elements of GCN & a traditional non-graph krigging method (i.e., GPs)?

2. Could the authors elaborate more on why this paper is specifically positioned in the temporal-spatial setting even though the proposed technique does not really have any new innovations for temporal modeling? I might have missed something important here.

3. It will be good to run some comparative experiments with a few representative graph convolutional GPs.

---

> ### Author Response · Authors · 2023-11-15
> **Thanks for your review! Response [1/2]**
>
> > **W1, Q2**: Intuition behind Spatio-Temporal Graph Convolution (STGC) Module.
>
> Thanks for raising this good question! We reply to this question as follows:
> 1. For Q2 (why the setting is spatio-temporal), we claim that for the $i$-th node $X_i^t$ obtained from timestamp $t$:
>     - if it aggregates information from other nodes $X_j^t$ under the same timestamp $t$, where $j \neq i$, then the **spatial** aspect will be included in the setting;
>     - if it obtains information from different timestamps, e.g., $t-1$, $t+1$, then the **temporal** aspect will be captured in the setting.
> 2. More details about STGC module:
>     - Firstly, we explain the differences between normal GCN (spatial only) and STGC (spatial + temporal).
>         - In Figure 3(a), for node-1 under timestamp $T_i$, normal GCN would only aggregate node-1 with node-2,3,5's (node-1's neighbors) information under **the same timestamp $T_i$**;
>         - Differently, STGC would aggregate not only **neighbor information within the same timestamp**, but also **information from each node's neighbors across time**.
>     - Then, we explain the design and rationale behind STGC module:
>         - **Forbid each node to aggregate information from itself (including the same and different timestamps)**:
>             - As introduced in the paper, the nodes can be divided into 2 groups, namely observed nodes (with sensors) and target nodes (no sensors). Observed nodes have sensor readings all the time, while target nodes can never have sensor readings. The essence of kriging is to learn the pattern only on observed nodes during training, and directly apply the learned pattern to target nodes for value estimation. Due to the fact that observed nodes not only have sensor readings but also have loss regulations, their kriging patterns can be easily learned, i.e., their learned representations/features can be quite good.
>             - With the above viewpoint in mind, if we allow self-loops, then the observed nodes would keep aggregating *good-quality* features, while the target nodes would keep aggregating *bad-quality* features from itself across time. Therefore, the pattern learned on observed nodes cannot suit target nodes well, and the performance would be suboptimal.
>             - Therefore, we (1) remove self-loop in the adjacency matrix (**verified in row 1&3 of Table 11**); and (2) do not utilize RNN-like modules to aggregate standard temporal information (**verified in row 1&2 of Table 11**). For example, in Figure 3, node-1 ($T_i$) will not receive information from node-1($T_i$) and node-1($T_{i+1}, T_{i-1}$).
>         - **Aggregate each node with its neighbors information (from the same timestamp)**: The intuition is the same as that of the normal GCN.
>         - **Aggregate each node with its neighbors information (from different timestamps)**: This design is less common yet effective (**verified in row 4-8 of Table 11**). We provide a few examples to explain the intuition as follows:
>             - Consider that we have road 1 and road 2, and the vehicles would drive from road 1 to road 2. Suppose at 8:00 am, a traffic jam occurs at road 2, and affected by road 2, road 1  gets crowded at 8:05 am. Therefore, the data distributions (e.g., vehicle speed data) of road 1 (8:05) have high correlations with those of road 2 (8:00).
>             - Similarily, suppose a group of vehicles drive on road 1 at 8:00 am, and the same group drive to road 2 at 8:05 am. The driving behavior of the same group might be uniform, thus, the data distributions (e.g., vehicle speed data) of road 2 (8:05) have high correlations with those of road 1 (8:00).
>     - Finally, the hyperparameter $m$ is used to define the scope, e.g., if $m=2$, then we consider each node at $T_i$ has high correlations with its neighbors at $\{T_{i-2}, T_{i-1}, T_i, T_{i+1}, T_{i+2}\}$, and would aggregate the former with the latter.

---

> > ### Author Response · Authors · 2023-11-15
> > **Response [2/2]**
> >
> > > **W2, Q1, Q3**: Discussion about Graph Convolutions with Gaussian Process.
> >
> > Many thanks for this comment! Our responses are as follows:
> > 1. You are right that "Gaussian processes were referred to as krigging in the past (in geo-statistics)", and **we have included it in our baselines, which is called OKriging**. However, this method requires GPS coordinates as inputs, and some datasets do not have such information, therefore, the results of OKriging are missing in some datasets.
> > 2. To the best of our knowledge, there is no existing GCN+GP method particulary designed for the kriging task, but we do agree it's a natural and good idea to combine GCN and GP for kriging. Therefore, we have followed your advice to include one of the latest GCN+GP methods published in ICLR'23 called GNN-GP (https://openreview.net/forum?id=flap0Bo6TK_), and conducted some experiments as follows:
> >     - We first clarify the differences between the data used in GNN-GP and our paper:
> >         - GNN-GP focuses on graph regression datasets like Chameleon to perform semi-supervised regression task. This dataset has **1 data sample**, 2,277 nodes and **3,132 input features**. The data is transformed from **$2,277 \times 3,132$** to **$2,277 \times 1$**.
> >         - For the dataset METR-LA, adopted in this paper, it has **6,872 test samples**, 207 nodes and **1 input features**. The data is transformed from **$207 \times 1$** to **$207 \times 1$**.
> >     - Then, we highlight the adaptions we've made to apply GNN-GP to our kriging task, and show the best MAE scores (after grid search) in the table below:
> >         - **Option 1**: Forward GNN-GP 6,872 times, the shape of input is $207 \times 1$ (speed features of current timestamp), the shape of output is $207 \times 1$.
> >         - **Option 2**: Forward GNN-GP 6,872 times, the shape of input is $207 \times 6,872$ (speed features from all timestamps), the shape of output is $207 \times 1$.
> >         - **Option 3**: Forward GNN-GP 1 time, the shape of input is $207 \times 6,872$ (speed features from all timestamps), the shape of output is $207 \times 6,872$.
> >         - **Option 4**: Forward GNN-GP 287 times, and process 24 samples together in each forward operation. This option is aligned with baselines and our KITS, i.e., in each data sample, we would use data from 24 consecutive timestamps for aggregating temporal information. The shape of input is $207 \times 24$ (speed features from 24 timestamps), the shape of output is $207 \times 24$.
> >     - We could observe that GNN-GP does not perform well in this task (we've already tuned the hyperparameters), e.g., it has the worst behavior, compared with all methods in Table 2 (e.g., the MAE score of our KITS is 6.1276). We discuss the possible reasons as follows:
> >         - GNN-GP originally requires 3,132 diverse input features, but there is only 1 input feature available in kriging. Although we can take data from different timestamps to enrich the input features, such input features are different from those used in GNN-GP.
> >         - Unlike the learnable parameters in baselines and KITS which can witness all data samples, and could learn and remember some intrinsic attributes of each node, the Gaussian kernels in GNN-GP cannot have such ability.
> >         - As mentioned in one of the baselines INCREASE (WWW'23, Section 2.2), the Gaussian assumption is quite restrictive, as the data used in kriging may not follow Gaussian distributions.
> > 3. Based on the responses above, we think GCN+GP is an interesting idea for kriging, but the current GCN+GP methods have not been well adapted for kriging, and we think how to design GCN+GP particularly for kriging can be regarded as a possible future direction.
> >
> > |Input shape|Output shape|#Forward|Time cost|MAE|
> > |-|-|-|-|-|
> > |$207 \times 1$|$207 \times 1$|6,872|890s|10.6211|
> > |$207 \times 6,872$|$207 \times 1$|6,872|893s|10.6214|
> > |$207 \times 6,872$|$207 \times 6,872$|1|3s|**10.5634**|
> > |$207 \times 24$|$207 \times 24$|287|40s|10.5935|
> >
> > > **W3**: Summarize GCN in the draft.
> >
> > We shall include some representative GNNs in related work.

---

> > > ### Comment · Reviewer_env1 · 2023-11-19
> > > **Re: Thank you for the detailed response**
> > >
> > > Thanks for the detailed response & the extra empirical work.
> > >
> > > All look good to me & I maintain my original support for this paper.
> > >
> > > --
> > >
> > > That being said, I do want to push back a little on the statement that "there is no existing GCN+GP method particulary designed for the kriging task". Could the authors discuss this paper on graph GP: https://arxiv.org/pdf/1809.04379
> > >
> > > It seems like it has not been used in the exact context here but I am curious if it can be repurposed for that?
> > >
> > > This question is meant for constructive feedback only (my support will not change) so please feel free to expand a detailed discussion as necessary.

---

> > > > ### Author Response · Authors · 2023-11-20
> > > > **Thanks for your follow-up comment and your recognition of our work!**
> > > >
> > > > > That being said, I do want to push back a little on the statement that "there is no existing GCN+GP method particulary designed for the kriging task". Could the authors discuss this paper on graph GP: https://arxiv.org/pdf/1809.04379
> > > > It seems like it has not been used in the exact context here but I am curious if it can be repurposed for that?
> > > > This question is meant for constructive feedback only (my support will not change) so please feel free to expand a detailed discussion as necessary.
> > > >
> > > > Thank you for your follow-up comment. Our responses are as follows.
> > > > - The suggested baseline (which we refer to as GGP) is designed originally for the semi-supervisied **node classification** task. It was adapted for the semi-supervised **node regression** task in the GNN-GP paper (ICLR'23). Therefore, this adapted method can also be used for the kriging task targeted in this paper. Note that we have included the discussion on the GNN-GP method in our previous response.
> > > > - We checked that the GGP code (its adapted version for the semi-supervised node regression task) is available in the github respository of GNN-GP (https://github.com/niuzehao/gnn-gp). Therefore, we used the code and conducted experiments (with Option 1 specified in our previous response) on the METR-LA dataset. The best MAE score we have obtained for GGP (by exploring different hyperparameters) is 10.5997. As a comparison, the MAE score of our KITS is 6.1276. According our experimental results, GGP, just like GNN-GP, does not perform well on spatio-temporal kriging task. The reasons that we have discussed for explaining why GNN-GP does not work well for the kriging task in our previous response should remain applicable for GGP.

---

> > > > > ### Comment · Reviewer_env1 · 2023-11-22
> > > > > **Thanks for the detailed response**
> > > > >
> > > > > Thank you for the response. Please consider including this brief discussion in the revised paper (either in a related work section or as a remark in the experiments) for thoroughness.
> > > > >
> > > > > As mentioned previously, I will vote for acceptance.

---

> > > > > > ### Author Response · Authors · 2023-11-22
> > > > > > **Thanks for your reply!**
> > > > > >
> > > > > > Thank you again for your recognition of our work! We will include the discussion in our experiments.

---

### Official Review · Reviewer_Ge1P · 2023-11-20

**Soundness:** 3 good
**Presentation:** 3 good
**Contribution:** 3 good
**Rating:** 6
**Confidence:** 2

**Summary:**

This paper addresses the challenge of inferring unobserved nodes using observed source nodes. While several inductive spatio-temporal kriging methods utilizing graph neural networks have been previously proposed, they often do not account for the discrepancies between the sparsity of training data and inference data (as the 'graph gap').

The author introduces an approach that adds virtual nodes into the training graph. This is done by 1) improve bad-learned features by finding similar nodes/feature fuse (Reference-based Feature Fusion module) 2) improve supervision signals by construct reliable pseudo labels for virtual nodes (Node-aware Cycle Regulation). Authors also show extensive experiments on 8 datasets with ablation study to support the method.

**Strengths:**

Admittedly, I am not a domain expert for inductive spatio-temporal kriging methods based on GNN, I do find this paper is:
* Well-written, and enjoyable to read.
* This looks like a novel method to address the sparsity gap between training graph and inference graph.

**Weaknesses:**

* My main concern is that the introduction of virtual nodes does not add additional information to the dataset. Consequently, my intuition is that this strategy is more effective when the dataset size is small. In the case of the benchmark datasets, which contain a limited number of spatial data points (ranging from a maximum of 883 to a minimum of 80), the approach seems advantageous. However, in scenarios where there is a larger denser spatial dataset, it's worth considering whether the issue of sparsity remains as significant. Would the gap in sparsity still present a challenge in extensive populated spatial data environments? But again, I am not a domain expert for inductive spatio-temporal kriging methods based on GNN, maybe this is indeed the usual dataset size for this line of work. I am happy to change my score if this gets justified.

* This paper employs a thresholded Gaussian kernel to construct the adjacency matrix A among nodes, setting the values to zero when distances exceed a certain threshold. Have the authors explored the possibility of using non-stationary kernels or neural network-based kernels as alternatives?

**Questions:**

* When aggregating spatio-temporal features from neighboring nodes, is it worth exploring some attention mechanism to gain more global information (especially with so many timesteps in the datasets)?

---

> ### Author Response · Authors · 2023-11-20
> **Thanks for your review!**
>
> > **W1**: The sparsity of deployed sensors.
>
> Thanks for this comment. Our responses are as follows:
>
> As written in the 1st paragraph of introduction ("Nevertheless, due to the **high cost of devices and maintenance expenses**, the actual **sparsely deployed sensors**..."), we target the kriging setting where the sensors are sparsely deployed (mainly for cost saving considerations). This setting has been studied in many existing kriging methods, including IGNNK (Wu et al., 2021a), LSJSTN (Hu et al., 2021) and INCREASE (Zheng et al., 2023). For this setting, the **graph gap issue** would remain valid. We further provide a few **practical application scenarios** for justifying the targetted setting.
>    1. The PEMS07 dataset comes from the Caltrans Performance Measurement System (PeMS) deployed in California (in use). It covers a large district, but the average node degree is 1.79 for the full graph (883 nodes), which means the graph is indeed locally sparse.
>    2. Similarily, the AQI-36 dataset is collected from an air quality monitoring application, where only 36 stations are installed for the whole Beijing City.
>    3. In our own business data of traffic, only 71 traffic sensors (to measure traffic flow) are sparsely installed to a large region, centered at Jinghu Amusement Park, Shaoxing, Zhejiang, China. The average node degree is less than 3.
>    4. In many water quality management systems, the sensors are sparsely installed. For example, the Rhein river in Europe only has 2-3 sensors in each city it flows through.
>    5. In a landslides monitoring application, there are only 7 landside tilt sensors installed in Yushan mountain in Taiwan.
>
> We are not aware of any existing studies/datasets/practical scenarios, which target the kriging task with desnsely distributed sensors. We believe the kriging with densely distributed sensors would naturally be an easier task than the one targeted in this paper since each node would have more neighbors to leverage for the kriging task.
>
> > **W2**: Non-stationary or neural network-based kernels.
>
> It's a very reasonable and interesting idea! When we worked on this paper, we did not come up with this idea, but we do agree it can potentially be  applied to this task. Particularly, we also wonder if the virtual nodes-related graph edges (in adjacency matrix) can be learnable instead of being randomly determined (in the current version). To keep consistency with baselines, we would still use Gaussian-thresholded static kernels in this paper, and consider your idea as a very good future direction. Thanks again for this precious comment!
>
> > **Q1**: Use attention mechanism for global spatio-temporal features aggregation.
>
> Thank you for your suggestion! Our understanding of the suggested idea is as follows - in case we misunderstood the idea, please kindly let us know. Suppose node-1's neighbors are node-2, 3, 4, and their data from 24 consecutive timestamps are available in a data sample. Your suggested idea is to use attention to aggregate node-1's features (in 1 of 24 timestamps) with node-2, 3, 4's features (from all 24 timestamps), so as to capture global dependency (in the temporal dimension).
>
> Our responses are as follows:
> - **Temporal scope (from local to global)**: We have studied the effects of the temporal scope, which we represent with a parameter $m$ (i.e., the information over $2 \times m + 1$ timestamps is aggregated). According to the results shown in Table 11, we can observe that larger $m$ (more global temporal information) cannot guarantee better performance.
> - **GCN v.s. GAT/Attention**: When we started this work, we also thought about which to use among GCN, GAT or standard Self-Attention (without adjacency matrix). We finally chose GCN, and we explain as follows:
>     - GAT and Self-Attention would firstly perform projection operations to project input features to $Q$, $K$ and $V$ in attention blocks. However, we think such projections cannot work well in spatio-temporal kriging. This is because in kriging, observed nodes would consistently have loss supervisions, while target nodes do not have loss supervisions all the time. Hence, the complex projections learned on observed nodes may not be suitable for target nodes during inference since it would get overfitting on observed nodes.
>     - We conduct experiments by replacing GCN (M-0 of Table 5) with GAT and Self-Attention under the same condition. We show the MAE scores on METR-LA dataset in the table below (we've already tuned the hyperparameters). We could see that attention-like methods cannot outperform GCN-like methods.
>
> |Method|MAE|
> |-|-|
> |GCN|**6.7597**|
> |GAT|7.3220|
> |Attention|8.2679|

---

> > ### Author Response · Authors · 2023-11-24
> > **Reminder**
> >
> > Dear Reviewer Ge1P,
> >
> > We sincerely appreciate the time and effort you have dedicated to reviewing our work. We have addressed the your questions and concerns above. We kindly request you to reconsider your score if you are satisfied with our justifications on the problem setting and datasets.
> >
> > By the way, please be reminded that the author response period for our submissions has been extended until the end of December 1st. We would really appreciate it if our next round of communication could leave time for us to resolve any of your remaining or new questions.
> >
> > Thank you so much for devoting time to improving our work!
> >
> > Regards,
> >
> > Authors

---

> > > ### Comment · Reviewer_Ge1P · 2023-11-25
> > >
> > > Thank you. I think most of my concerns are addressed. I raised my score to 6 and kept my confidence as 2. The reason I cannot raise my score/confidence higher is because I am not a domain expert for inductive spatio-temporal kriging methods based on GNN so I cannot rate based on the impact of this topic, but I do feel this paper is enjoyable to read and I like the idea in the paper.

---

> > > > ### Author Response · Authors · 2023-11-26
> > > > **Thanks for your quick reply!**
> > > >
> > > > We are pleased to hear that your concerns are addressed and you like our idea. Thanks again for your nice review and your recognition of our work!

---

### Author Response · Authors · 2023-11-28
**Request for discussion on the graph gap issue**

Dear PC, AC and Reviewers,

First of all, thank you very much for the time you have devoted to reviewing our submission: KITS: Inductive Spatio-Temporal Kriging with Increment Training Strategy and providing us the constructive comments. Below, we would like to summarize the reviews on our paper and then request for more discussions on the graph gap issue that we aim to solve in our paper.

**(1) Summary of reviews (from ICLR)**: Up till the present moment, we have received 3 reviews, with 2x Score 6 (env1, Ge1P) and 1x Score 5 (FRYk). We are encouraged that you find our idea **novel (all reviewers)**, refreshingly interesting (env1), appealing (FRYk, Ge1P), the paper is well-written (FRYk, Ge1P) and enjoyable to read (Ge1P), the experiments are sufficiently extensive (env1), and good empirical performance is achieved (env1, FRYk). Particularly, Reviewer env1 recognizes the significance of the proposed "graph gap" issue, and thinks the problem being addressed in this paper is practically significant.

**(2) Summary of reviews from a previous conference**: As Reviewer FRYk has mentioned, he/she was one of the reviewers for our paper when it was submitted to a previous conference. For that submission, we got **3x "Weak Accepts (Score 6)"** and 1x "Borderline Reject (Score 4)" (which was recommended by Reviewer FRYk). Particularly, **all 4 reviewers (including Reviewer FRYk) find our methodology novel**. For more details, please refer to our point-to-point responses to these review comments in our response titled "Response [1/3]" and "Response [2/3]" to Reviewer FRYk.

**(3) Request for more discussions on the "graph gap" issue**. Reviewer FRYk insists on his opinion that the graph gap issue is not significant as indicated in his/her latest comment as follows.
> As discussed several times, I believe the so-called "graph gap" issue is not a thing. Removing outgoing edges from the virtual nodes would effectively remove any mismatch between the training and target graph at inference time.

However, we believe this is an unfair comment, for which we explain as follows. **First**, existing studies (and their training strategies), including IGNNK (Wu et al., 2021a), LSJSTN (Hu et al., 2021) and INCREASE (Zheng et al., 2023), all suffer from the "graph gap" issue - this issue has largely remained unsolved. **Second**, simple treatment for the issue would not work well for our problem setting (with sparsely distributed sensors). As we have verified in our experiments (Appendix H.8), the simple treatment as mentioned by Reviewer FRYk does not work effectively or efficiently for our problem setting, for which we have provided the reasons in our previous responses to his/her earlier comments. We believe this could be the major reason why this simple treatment has not been adopted by any of the existing studies.

In fact, the significance of the graph gap issue has been well recognized by other reviewers: (1) Reviewer env1 considers the graph gap issue as practically significant by saying **"The gap between the train and test graphs in the inductive setting of node prediction is always a fundamental issue, which needs an in-depth treatment."**; (2) Reviewer Ge1P subscribes to our view that the **"graph gap" exists under our setting**; and (3) One of the reviewers from previous conference thinks **this issue is reasonable**.

We believe we have made it clear about the graph gap issue: including (1) what it is; (2) why existing studies suffer from it; (3) how our method would largely mitigate the issue; and (4) why simple treatments would not work for our problem setting. Given that Reviewer FRYk still insists on his opinion that the graph gap issue is not significant, we would like to sincerely request for some more discussions among the reviewers on the "graph gap" issue (i.e., its significance). If there are any further clarifications required from our side, we would be more than happy to respond further.

Thank you very much!

Yours sincerely,

Authors

---

> ### Comment · Reviewer_FRYk · 2023-11-28
>
> Dear all,
>
> I believe I have discussed my opinion on this aspect at length. However some clarifications:
>
> 1. The experiment in Appendix H.8 simply shows that incremental reconstruction does not provide a performance improvement, but does not show that this so-called "graph gap" is the issue causing poor performance and that the proposed method performs well because it's addressing this specific problem. Actually, the experiment confirms that the improvement might not come from removing the gap.
> 2. Again, I'm not saying the training methodology proposed in KITS is not effective. My opinion is that the effectiveness of the method does not come from solving a mismatch in the graph used for training and inference as again it could *easily* be avoided in these settings. The introduced data augmentations are well aligned with the kriging task and support inductive learning, i.e., extrapolation to new nodes and unseen locations, and I believe that this is what could justify the improvement (this should be empirically validated). However, I believe that identifying the "graph gap" (as formulated in the paper) as the main issue of the existing methods is misleading (again it can be avoided very easily).
> 3. I don't think that any of my comments is "unfair".
>
> That being said, as I already mentioned I believe the paper has merits.

---

> > ### Author Response · Authors · 2023-11-29
> > **Responses [1/2]**
> >
> > For ease of discussion, we copy Table 17 from Appendix H.8 as below. Note that "Single" is the strategy recommended by Reviewer FRYk, "Decrement" is the strategy adopted by baselines, and "Increment" is our proposed strategy.
> >
> > |Strategy|\#Training nodes|\#Inference nodes|\#Iterations|METR-LA (207)|SEA-LOOP (323)|PEMS07 (883)|AQI (437)|
> > |-|-|-|-|-|-|-|-|
> > |Single (suggested) |x|x+1|x|7.3560|4.9766|146.2889|28.2472|
> > |Decrement (existing) |x|2x|1|6.7597|4.5732|86.7923|17.5867|
> > |Increment (ours) |2x|2x|1|**6.3754**|**4.3688**|**82.9259**|**16.5797**|
> >
> > The model with "Increment" strategy only consists of a naive GCN model and the proposed Increment training strategy (Section 3.2), and the models with "Single" and "Decrement" strategies adopt the same GCN model with the suggested strategy and the baselines' training strategy, respectively.
> >
> > Next, we would like to respond to Reviewer FRYk's follow-up comments.
> >
> > > The experiment in Appendix H.8 simply shows that incremental reconstruction does not provide a performance improvement, but does not show that this so-called "graph gap" is the issue causing poor performance and that the proposed method performs well because it's addressing this specific problem. Actually, the experiment confirms that the improvement might not come from removing the gap.
> >
> > **(1) Decrement v.s. Increment**: We first compare Decrement and Increment in two aspects, namely, (1) Decrement has the graph gap issue while Increment does not have the graph gap issue; and (2) both have their inference graphs involve all unobserved nodes collectively. The results show that Increment works better than Decrement, which further implies that **removing the graph gap helps to improve the performance** because Decrement and Increment differ in the graph gap issue aspect only.
> >
> > **(2) Increment v.s. Single**: We then compare Increment and Single: (1) Both do not have the graph gap issue; and (2) Increment has its inference graph (with 2x nodes) involving all unobserved nodes collectively while Single has its inference graph (with (x+1) nodes) involving one unobserved node. The results show that Single works worse than Increment. **These results support that it is necessary to collectively involve all targeted/unobserved nodes in the inference graph, but this would naturally impose the graph gap issue**, i.e., the gap between the training graph (without unobserved nodes) and the inference graph (with unbserved nodes). The Single strategy involves one unobserved node in the inference graph each time (for removing the graph gap straightforwardly), but it fails to collectively involve all unobserved nodes in the inference graph. As a result, Single does not work well.
> >
> > **(3) Decrement v.s. Single**: We lastly compare Decrement and Single. Decrement and Single differ in both of the aforementioned aspects, namely (1) Decrement has the graph gap issue while Single does not and (2) Decrement has the inference graph (with 2x nodes) involving all unobserved nodes collectively while Single has its inference graph (with (x+1) nodes) involving one unobserved node. The results show that Single works worse than Decrement. This is because Single is better in that it does not have the graph gap issue, but it is worse in that it fails to involve all unobserved nodes collectively in the inference graph.
> >
> > Our conclusion based on the results is that **removing the graph gap issue (while involving all unobserved nodes in an inference graph) would help to improve the performance**. However, this is not an easy task, which is in contrast to Reviewer FRYk's interpretation. Our KITS achieves this goal with our newly proposed Increment training strategy while the suggested Single strategy does not.

---

> > > ### Author Response · Authors · 2023-11-29
> > > **Responses [2/2]**
> > >
> > > > Again, I'm not saying the training methodology proposed in KITS is not effective. My opinion is that the effectiveness of the method does not come from solving a mismatch in the graph used for training and inference as again it could easily be avoided in these settings. The introduced data augmentations are well aligned with the kriging task and support inductive learning, i.e., extrapolation to new nodes and unseen locations, and I believe that this is what could justify the improvement (this should be empirically validated). However, I believe that identifying the "graph gap" (as formulated in the paper) as the main issue of the existing methods is misleading (again it can be avoided very easily).
> > >
> > > We thank Reviewer FRYk for his/her recognition of our proposed methodology and the suggested way of explaining the merits of our methodology. We believe both our way (i.e., attributing the improvement to the fact that our method mitigates the graph gap) and the suggested way by Reviewer FRYk (i.e., attributing the improvement to the fact our method is better aligned the nature of inductive kriging than decrement strategies) are reasonable. We have the considerations: (1) for our way, the experimental results (i.e., those in Appendix H.8) can provide some verification; and (2) for the suggested way, while it is intuitive, there is no clear clue of validating the explanation. Having said that, we can include the suggested way of explaining the merits of our method by Reviewer FRYk in our paper as additional insights of our method if Reviewer FRYk think this would help to resolve the presentation issue we are discussing.

---

> > > > ### Comment · Reviewer_FRYk · 2023-11-30
> > > >
> > > > The fact that what you call "single strategy" underperforms simply means that in the considered setting using a dense graph for inference can help. However, this can be achieved in many ways and has nothing to do with the "graph gap". Again, I feel like I'm repeating myself, but you are identifying with this "graph gap" as the core of the issue here, but as already mentioned it is a **completely avoidable** issue in virtual sensing as described above.
> > > >
> > > > Then, the fact that sparsely connected graphs might lead to poor results has nothing to do with "the graph gap" and is an issue that can be addressed in many different ways, e.g., rewiring the graph and/or with data augmentations. That is why I am suggesting a different framing.

---

### Meta-Review · Area_Chair_QLb8 · 2023-12-04

**Metareview:**

While some reviewers found the ideas interesting, the highest scores were two scores of Weak Accept (6), which were not provided with strong confidence. The Weak Reject score (5) was provided with higher confidence, and several reviewers engaged heavily in the discussion, including this one. Issues raised include incorrect interpretation of results, and too small of a graph to draw such conclusions.

**Justification For Why Not Higher Score:**

Borderline paper with no score higher than a 6, even after long reviewer discussions with authors.

**Justification For Why Not Lower Score:**

N/A

---

### Decision · Program_Chairs · 2024-01-16

Reject